

# Northern Hemisphere continental winter warming following the 1991 Mt. Pinatubo eruption: Reconciling models and observations

Lorenzo M. Polvani[1,2], Antara Banerjee[1], and Anja Schmidt[3,4]

[1]Department of Applied Physics and Applied Mathematics and Department of Earth and Environmental Sciences,Columbia University, New York, NY 10027, USA
[2]Lamont Doherty Earth Observatory, Columbia University, Palisades, NY 10964, USA
[3]Department of Chemistry, University of Cambridge, Lensfield Road, Cambridge CB2 1EW, UK
[4]Department of Geography, University of Cambridge, Downing Place, Cambridge CB2 3EN, UK

*Correspondence to:* Lorenzo M. Polvani (LMP@COLUMBIA.EDU)

**Abstract.** It has been suggested, and is widely believed, that the anomalous surface warming observed over the Northern Hemisphere continents in the winter following the 1991 eruption of Mt. Pinatubo was, in fact, caused by that eruption, via a stratospheric pathway that involves a strengthening of the polar vortex. However, most studies that have examined multiple, state-of-the-art, coupled climate models report that, in the ensemble mean, the models do not show winter warming after the

Mt. Pinatubo eruption. This lack of surface warming in the multi-model mean, concomitant with a lack of strengthening of the polar vortex, is often interpreted as a failure of the models to reproduce the observations. In this paper we show that this interpretation is erroneous, as averaging many simulations from different models, or from the same model, is not expected to yield surface anomalies similar to the observed ones, even if the models were highly accurate, owing to the presence of strong internal variability.

We here analyze three large ensembles of state-of-the-art, coupled climate model simulations and show that, in all three, many individual ensemble members are able to produce post-Pinatubo surface warming in winter that is comparable to the observed one. This establishes that current-generation climate models are perfectly capable of reproducing the observed surface post-eruption warming. We also confirm the bulk of previous studies, and show that the surface anomaly is not statistically different from zero when *averaged* across ensembles of simulations, which we interpret as the simple fact that the volcanic

impact on continental winter temperatures is tiny compared to internal variability.

We also examine the stratospheric pathway and, again confirming previous work, show that any strengthening of the polar vortex caused by the Mt. Pinatubo eruption is likely to be very small (of the order of a few m/s at best). Such minuscule anomalies of the stratospheric circulation are completely overwhelmed by the tropospheric variability at mid-latitudes, which is known to be very large: this explains the lack of surface winter warming in the ensemble means.

In summary, our analysis and interpretation offers compelling new evidence that the observed warming of the Northern Hemisphere continents in the winter 1991-1992 was very likely unrelated to the 1991 Mt. Pinatubo eruption.



## 1 Introduction

Large, low-latitude volcanic eruptions produce considerable, albeit short lived, natural perturbations to the radiative forcing of the Earth's climate, and thus offer unique opportunities to probe its dynamics. With an estimated peak aerosol loading of 30 Tg (McCormick and Veiga, 1992), the eruption of Mt. Pinatubo in June 1991 was the largest to occur since the advent satellite

observation and, in fact, the second largest over the entire 20th century (after the 1912 Novarupta eruption). Moreover, in terms of dust veil index (Robock, 2000) and stratospheric optical depth (Sato et al., 1993) it stands unrivaled all the way back to the historic eruption of Mt. Krakatau in 1883, and is therefore the premier candidate for understanding how volcanic aerosols affect the climate system.

After the initial cataclysmic eruption of June 14-15 1991, the aerosol cloud from Mt. Pinatubo spread rapidly and encircled
the globe in a mere 22 days (Bluth et al., 1992) filling the entire tropical belt, both north and south of the Equator, in a couple of months (McCormick and Veiga, 1992) and then spreading to higher latitudes in subsequent months (Long and Stowe, 1994). Since volcanic aerosols are strong scatterers of incoming solar radiation, they act to cool the troposphere and the Earth's surface. By September 1992, the global lower troposphere had cooled by -0.5°C (Dutton and Christy, 1992), with an even larger cooling of -0.7°C in the Northern Hemisphere (NH). Such large cooling values are comparable to the estimates for the
epochal Tambora eruption of 1815 (McCormick et al., 1995).

In the context of such widespread cooling, the surface temperature over the NH continents happened to be anomalously warm in the winter immediately following the Mt. Pinatubo eruption (Robock, 2002). In a series of papers, Groisman (1992), Robock and Mao (1992), and later Robock and Mao (1995) and Kelly et al. (1996), argued that continental winter warming also occurred following several other eruptions since 1850, and suggested that the winter NH warming was actually *caused* by
the volcanic eruptions. Further observational evidence was offered by Shindell et al. (2004), who expanded the set to a dozen large, low-latitude eruptions, going back to the year 1600. Their additional evidence, however, includes some perplexing facts. For instance, they show that the continental winter warming following both the 1883 Krakatau and the 1815 Mt. Tambora eruptions is, apparently, much smaller than the one following the 1982 El Chichón eruption (see Figure 1 of Shindell et al., 2004): this is difficult to reconcile with the narrative that volcanoes are the major cause of the NH continental winter warming,
since those two earlier eruptions are larger in magnitude than the later one.

Part of the widespread belief in the existence of a causal link between low-latitude volcanic eruptions and winter warming over the NH continents stems from the fact that a mechanism has been proposed to explain that link. Graf et al. (1993), on the basis of highly[1] idealized numerical experiments, followed by the observational studies of Kodera (1994) and Perlwitz and Graf (1995), and further numerical studies by Kirchner et al. (1999), Stenchikov et al. (2002) and many others thereafter, have
advocated for the existence of what we will refer to as a "stratospheric pathway" causally linking low-latitude eruptions in summer with mid-latitude surface warming the following winter. The starting point for this mechanism is the well known fact that sulfate aerosols of volcanic origin are also strong absorbers of infrared radiation: hence powerful, low-latitude eruptions

---

[1]Their model was run in perpetual January configuration, with prescribed sea surface temperatures and sea ice concentrations, forced with an "externally computed" heating rate, but without interactive aerosols or ozone chemistry modules.



that are able to penetrate sufficiently high into the atmosphere can cause a strong *warming* of the tropical lower stratosphere, in addition to the tropospheric and surface cooling mentioned above. In the case of Mt. Pinatubo a 2-3°C warming[2] of the tropical lower stratosphere was seen in radiosonde observations (Randel, 2010), in agreement with multiple reanalyses (Fujiwara et al., 2015). Such a perturbation increases the equator-to-pole temperature gradient in the stratosphere, notably in winter, and induces

a strengthening of the stratospheric polar vortex. The stronger polar vortex, it is claimed, then causes a positive phase of the North Atlantic Oscillation (or the Northern Annual Mode), which finally results in warmer surface temperatures over the NH continents, notably over Eurasia.

In spite of its simplicity, this proposed mechanism remains unconvincing because it has yet to be properly quantified. For instance, one could ask: *how large* is the polar vortex acceleration caused by an eruption comparable to the one of Mt.

Pinatubo in 1991? A recent study (Bittner et al., 2016), using a very large ensemble of runs with a well-tested stratosphere-resolving model, suggests a polar vortex acceleration possibly as large as 2 m/s at 10 hPa around 60N, but also reports that even 100 model runs are insufficient to establish that fact at the 99% confidence level and if one lowers the level to 95% more than 60 runs are needed for a statistically significant 2 m/s acceleration of the polar vortex (see their Figure 2a). Moreover, the large internal variability associated with the North Atlantic Oscillation can easily overwhelm the surface effects of such a

small stratospheric perturbation, as it even confounds the forced signal from increasing greenhouse gases over an entire 50-year period (see, for instance, Deser et al., 2017).

In fact, the original stratospheric pathway mechanism has been called into question, even by its original proponents. Stenchikov et al. (2002) suggested that the stratospheric pathway may be part of a more complex mechanism and, on the basis of results from a single model, proposed that an additional tropospheric pathway may be equally important. In addi-

tion, Graf et al. (2007) reported that observations actually show *increased* planetary wave activity in the winter following the Mt. Pinatubo eruption, which is clearly at odds with the claim of a stronger polar vortex that winter causing the NH surface warming, and completely invalidates the original mechanism. Thus, they suggest "that the climate effects of volcanic eruptions are *not* being explained by the excitation of inherent zonal mean variability modes such as Strong Polar Vortex or Northern Annular Mode, but rather is another mode that possibly reflects upon the North Atlantic Oscillation" (Graf et al., 2007).

Furthermore, one can find in the literature many modeling studies whose findings are often diametrically opposite to each other. We will not exhaustively cite all previous papers, but simply limit ourselves to highlighting a few key studies to illustrate the contradictory claims that can be found in the peer-reviewed literature. Let us start by summarizing the findings of Driscoll et al. (2012), who analyzed 13 models from the Climate Model Intercomparison Project, Phase 5 (CMIP5). These models were specifically selected so as to have at least two ensemble members available. Comparing the average across all the models, as

well as the averages across all the members of the each model, they concluded that "none of the models manage to simulate a sufficiently strong dynamical response," given the absence of NH continental warming following the Mt. Pinatubo eruption in the model averages. Their study confirms the earlier conclusion reached with the CMIP3 models (Stenchikov et al., 2006), and many other studies (e.g. Thomas et al., 2009; Marshall et al., 2009; Bittner, 2015; Wunderlich and Mitchell, 2017).

---

[2]At levels close to 20 km, taking the one-year mean after the eruption minus the mean of the preceding three years.





Against this body of evidence, analyzing two version of the NASA/GISS model, Shindell et al. (2004) have claimed that "driven by solar heating induced by the stratospheric aerosols, these models produce enhanced westerlies from the lower stratosphere all the way to the surface" and a significant wintertime warming over the NH continents, in agreement with Graf et al. (1993) and Kirchner et al. (1999), who also claimed that climate models are able to simulate the continental winter

warming following the Mt. Pinatubo eruption via the stratospheric pathway. In fact, Shindell et al. (2004) concluded that their results "provide a further strong indication of the critical role of the stratosphere in the dynamic response to external forcing," with a suggestion that a well resolved stratosphere is crucial for capturing the NH winter warming that would be caused by volcanic eruptions. That suggestion, however, would seem soundly refuted by the evidence presented in Charlton-Perez et al. (2013), who separately analyzed models with and without a well-resolved stratosphere, and showed no difference between the

two sets in the forced response of the polar vortex in the winter following volcanic eruptions.

And lastly, Zambri and Robock (2016) reanalyzed the CMIP5 models using a different methodology. Averaging only the largest eruptions, and only the first winter after those eruptions, they concluded that "most models do produce a winter warming signal, with warmer temperatures over NH continents and a stronger polar vortex in the lower stratosphere," directly contradicting Driscoll et al. (2012).

It is in the context of such multiple inconsistent claims, that our paper aims to answer two questions:

1. Are current-generation climate models able to simulate the continental winter warming in the NH following the 1991 Mt. Pinatubo eruption?

2. If so, does the stratospheric pathway proposed by Robock and Mao (1992) and Graf et al. (1993) play any role in simulating that warming?

Analyzing large ensembles of model integrations from three different state-of-the-art coupled climate models over the historical period, we show below that (1) models *are* perfectly capable of simulating NH continental warming in the winter following the Mt. Pinatubo eruption, but (2) the stratospheric pathway – and, more importantly, the Mt. Pinatubo eruption itself – very likely played *no significant role* in the occurrence of that warming.

## 2   Methods

### 2.1   The models

Three large ensembles of integrations with state-of-the-art, comprehensive climate models are analyzed in here. All our models include atmosphere, land, ocean and sea-ice components, fully coupled [3] to accurately simulate the climate system response to the Mt. Pinatubo eruption. Here are, in brief, the specifications of our three models: WACCM4, CAM5-LE, CanESM2

  – WACCM4 is the Whole Atmosphere Community Climate Model, Version 4, developed by the Community Earth System

Model (CESM) Project. WACCM4 is a high-top model, with a lid at 140 km and 66 vertical levels, and a horizontal

---

[3]Note that was mostly not the case in the earlier studies. Neither Graf et al. (1993), not Kirchner et al. (1999), nor Stenchikov et al. (2002), nor Shindell et al. (2004) used fully coupled climate models.



resolution of ∼2°. Its climate over the 20th century has been thoroughly evaluated by Marsh et al. (2013), where further details about this model may be found. We emphasize that WACCM4 also includes interactive stratosphere ozone chemistry and, therefore, has the most realistic representation of stratospheric dynamics and chemistry of the three models analyzed here.

– CAM5-LE was also developed under the CESM project, with ocean and sea ice components similar to those of WACCM4. However, the atmospheric component of CAM5-LE is very different: it is a low-top model with only 30 vertical levels but with a higher horizontal resolution (∼1°) and, most importantly, employs very different physical parameterizations than those in WACCM4 (Neale et al., 2010) and, in fact, has a considerably different climate sensitivity (Gettelman et al., 2013). CAM5-LE has been at the heart of the CESM Large Ensemble Project (see Kay et al., 2015, for details) and its
performance, therefore, has been thoroughly tested in dozens of studies which have analyzed its output.

  – CanESM2 is the second-generation Canadian Earth System Model, developed at the Canadian Centre for Climate Modeling and Analysis (CCCma). The atmospheric component of CanESM2 is a spectral model with an approximate horizontal resolution of 2.8°and with 35 unevenly spaced vertical levels and a model top near 0.1 hPa. For more details the reader may consult von Salzen et al. (2013). Again, this is a well-tested model which has contributed a whole suite of runs to
the CMIP5 project, and it has been widely used in many climate studies (e.g. Swart et al., 2015).

We note that all three models were previously used to study the climatic effects of volcanic eruptions (English et al., 2013; Lehner et al., 2016; Gagné et al., 2017). More importantly, for all three models we have available a *large ensemble* of integrations which cover the second half of the 20th century. For these integrations, the models include all known natural and anthropogenic forcings, as per the so-called "historical" specifications of the CMIP5 protocol (Taylor et al., 2012). Specif-
ically, we have analyzed 13 runs with WACCM4, 42 runs with CAM5-LE, and 50 runs with CanESM2. We stress that the model forcings are *identical for all members* of the same ensemble. The differences among members of the same ensemble arise uniquely from minuscule perturbations imposed on the models' atmospheric initial conditions: the differences allow us to explore the internal variability of the system which, in many cases, can be much larger than the response to an external forcing, be it natural or anthropogenic. The reader is referred to Deser et al. (2012) for the seminal exposition of this methodology.

**2.2 The analysis**

We here discuss three key methodological choices we made in designing the best strategy to determine whether current-generation climate models are able to capture the wintertime NH continental warming following volcanic eruptions.

  1. *Choice of eruption.* Although the model runs available to us cover the 1963 Agung and 1982 El Chichón eruption, we will here focus solely on the 1991 eruption of Mt. Pinatubo, in view of the following. First, as already noted, that eruption
is the best observed of all known eruptions, and thus offers the best opportunity to contrasting models and observations. Second, one can easily argue that every eruption is unique: for instance, while the aerosol cloud from Pinatubo spread out in both hemispheres, the one of Mt. Agung spread primarily into the Southern Hemisphere (Viebrock and Flowers, 1968).





So, combining these seems inappropriate. Third, and most importantly: since we are seeking to isolate and quantify the *forced response* to volcanic eruptions, it make no sense to average eruptions of different magnitudes. This would be tantamount to trying to estimate the Earth's climate sensitivity by averaging together $2\times CO_2$ and $4\times CO_2$ model runs. Other recent studies have also argued against averaging stronger and weaker eruptions when seeking to isolate their climatic impacts (Bittner et al., 2016; Zambri and Robock, 2016).

2. *Choice of winters.* We will here analyze only the *first* winter following the June 1991 eruption, i.e. the three month period from December 1991 to February 1992. Many (if not most) of the earlier studies assumed that the effect of volcanic eruptions can be felt for several years, and averaged together the first and second winters after each eruption. We see no cogent reason for doing so: the Mt. Pinatubo volcanic aerosols were removed from the atmosphere with an e-folding timescale of about 12-months (Barnes and Hofmann, 1997), so that the aerosol optical depth in January 1993 is much smaller than in January 1992 (see also Long and Stowe, 1994). Furthermore, if indeed the stratospheric pathway is crucial to carrying the response down to the surface at higher latitudes, it is difficult to imagine what memory the stratosphere would posses to remember in the winter of 1993 an eruption that occurred in June 1991. The recent study of Zambri and Robock (2016) also argues that only the first winter should be used, since "averaging the first two winters after each eruption may have had a damping effect."

3. *Choice of reference period.* For this, we follow the methodology of Driscoll et al. (2012) and define the winter-time anomalies after the Mt. Pinatubo eruption as the difference between 1991/1992 winter and the mean of the winters in the 1985-1990 reference period. While this need not be the best way to quantify the post-eruption anomalies, we nonetheless adopt it in order to be consistent with recent studies who analyzed models similar to ours (Bittner et al., 2016; Zambri and Robock, 2016). As we will show below, our conclusions differ significantly from those of previous studies, and we want to make it clear that the choice of reference period is not at the root of those differences.

In summary then: for all quantities in all figures below (except Fig. 2) we will be showing and discussing anomalies defined as the difference between the first winter following the June 1991 Mt. Pinatubo eruption and the reference period defined in Driscoll et al. (2012). We will refer to these as the "post-Pinatubo anomalies," or just "the anomalies" for short and, for simplicity, denote them with a prime (e.g. $T_s'$ for the surface temperature anomalies).

## 3   Can climate models simulate the observed NH continental warming following the 1991 Mt. Pinatubo eruption?

It is useful to start by recalling what the observed wintertime, post-Pinatubo, surface temperature anomalies over the NH continents actually look like. They are shown in Fig. 1, from four different datasets: two observational ones, GISSTEMP (Hansen et al., 2010) and HadCRUT4 (Morice et al., 2012), and two reanalyses, NCEP/NCAR (Kalnay et al., 1996) and ERA-Interim (Dee et al., 2011). Note the excellent agreement between these four data products, which all show warming over both North America and the Eurasian continent. The fact that *both* continental masses were anomalously warm, is of relevance for the stratospheric pathway mechanism to be discussed in the next section. These anomalies are also in excellent agreement with





the lower tropospheric temperature anomalies from the Microwave Sounding Unit, Channel 2 (MSU2) satellite observations shown by Robock (2002), albeit for a slightly different reference period.

We now turn to analyzing the models. Before showing the simulated surface temperatures, however, we wish to illustrate the models' response to the Mt. Pinatubo eruption in the stratosphere, as the warming of the tropical stratosphere is an essential

component of the stratospheric pathway mechanism. The global top-of-the-atmosphere (TOA) net outgoing shortwave radiation anomalies are shown in the top row of Fig. 2, for WACCM4, CAM5-LE, and CanESM2, from left to right. These panels may be contrasted directly with Fig. 2 of Driscoll et al. (2012), as they demonstrate that our three models are comparable to most CMIP5 models.

The resulting warming of the tropical lower stratosphere (30S-30N) is shown in the bottom row of Fig. 2. The ERA-Interim

reanalyses are also shown for comparison (black curves in each panel). While the CanESM2 model appears to be in good agreement with the observations, both WACCM4 and CAM5-LE greatly overestimate the post-eruption warming in the lower stratosphere. Reanalyses show an anomaly of roughly 2°C , but those models show ensemble mean anomalies closer to 6 and 9 °C , respectively. This is not exceptional, as Driscoll et al. (2012) reports that most CMIP5 models simulate a much stronger anomaly than was observed (see their Fig. 3). The interesting point, however, is that we will be turning this model

bias to our advantage: as will become clear below, the fact that the WACCM4 model, in particular, simulates a stronger than observed warming of the tropical lower stratosphere after the Mt. Pinatubo eruption will greatly strengthen our interpretation and conclusion.

With this is mind, we now proceed to examine the surface temperature anomalies simulated by our three models following the Mt. Pinatubo eruption, shown in Fig. 3. It is important to keep in mind that for each ensemble member the post-

Pinatubo anomalies arise from two distinct sources: the external forcing and internal variability. The former is computed by averaging together all the members of each ensemble, as that procedure eliminates the internal variability. For WACCM4, CAM5-LE and CanESM2, the left column of Fig. 3 shows that forced response. It is abundantly clear that in the winter following the Mt. Pinatubo eruption, the models show *no statistically significant response* in NH continental surface temperatures.

We stress that this result is in agreement with most of the literature on this subject, notably the multi-model studies with the

CMIP3 and CMIP5 models (Stenchikov et al., 2002; Driscoll et al., 2012; Wunderlich and Mitchell, 2017), which have shown that the forced post-Pinatubo anomalies in CMIP-class models are not statistically significant. Moreover, it has been validated with an even larger ensemble size: Bittner (2015), employing a fully-coupled stratosphere resolving model, concluded that after Mt. Pinatubo "the continental winter warming over Northern Europe and Siberia is not significantly different from zero even with 100 ensemble members" (as shown in Fig. 6.4 of that doctoral dissertation).

However, and this is perhaps the key point of our paper: from the fact that the ensemble mean (i.e. the forced) anomalies are not significant, it is *erroneous* to conclude that the models are unable to simulate the NH continental winter warming following the eruption. Recall that the observed anomalies are not expected to resemble the ensemble mean of any set of simulation, as internal variability is superimposed to any forced response in the observations. The correct question to ask is: do any individual simulations resemble the observations? Or, more precisely: do the observed anomalies fall within the range, over the ensemble,

of the simulated anomalies? The answer to that question is a resounding yes, as we show next.





Since that answer crucially depends on the range of anomalies that any one model is able to simulate, we start by illustrating that range. In the middle column of Fig. 3 we show the extreme members, i.e. the members with the largest warming anomalies, for each of the three models we have analyzed. Noting that the color-bar is identical to the one in Fig. 1, it is clear that the models are able to simulate much stonger warming anomalies than the observed ones. Even more: different ensemble members

of the same models, with *an identical volcanic forcing*, are able to simulate equally strong *cooling* over the northern continents, as shown in the right column of Fig. 3, where the coldest members can be seen. The point of this figure is to illustrate how large the internal variability is (in these models), and how tiny the forced response is in comparison. For completeness, the surface temperature anomalies for each member of each ensemble are shown in supplementary Figs. S1-S3.

We quantify the relative magnitude of the forced response and the internal variability in Fig. 5 with box and whisker plots

for the quantity $T'_s$, defined as the surface temperature anomaly averaged over the landmasses in the region (40-70N, 0-150W), roughly corresponding to the Eurasian continent. First note that the mean of each ensemble is very near zero (a few tens of degrees at most, and not statistically significant), confirming the results of many previous studies that the forced response in the NH midlatitudes in the winter following the Mt. Pinatubo eruption is basically non-existent in the models. Second, the models are in reasonably good agreement about the internal variability, showing a warming/cooling range of 2 to 4°C on each side of

zero, which is much larger than the forced response. Third, and most importantly: the reanalysis (red dot) falls well within the simulated range, indicating that the models are perfectly capable of capturing the post-Pinatubo winter anomalies in the NH.

## 4   Does the stratospheric pathway play a role in simulating the NH winter warming following the Pinatubo eruption?

Having established that our three models are able to simulate the observed NH continental warming after the Mt. Pinatubo erup-
tion, we now turn to examining the stratospheric pathway mechanism proposed by Robock and Mao (1992), Graf et al. (1993)

and others. In a nutshell, that mechanism involves two steps: (1) a strengthening of the stratospheric polar vortex caused by the enhanced equator-to-pole lower stratospheric temperature gradient following the Mt. Pinatubo eruption which, in turn, causes (2) an anomalous atmospheric circulation resulting in a warming anomaly over the Eurasian continent.

To carefully investigate the existence of a possible stratospheric pathway, we will limit ourselves to the WACCM4 model, as the other two do not have an accurate representation of the stratosphere and, more importantly, of its variability. We recognize

that 13 members may perhaps not qualify as a "large" ensemble but, as we will show, the results presented below are in excellent agreement with those of Bittner et al. (2016) who used a much larger[4] 100-member ensemble.

Now, to quantify the strength of the polar vortex we compute the quantity $U'_{10}$, defined as the anomaly in the zonal mean, zonal wind at 10 hPa and 60N. This quantity is widely used for the detection of stratospheric sudden warmings (see, e.g., Charlton and Polvani, 2007; Butler and Gerber, 2018). To quantify the meridional lower stratospheric temperature gradient we

compute the quantity $\nabla T'_{50}$, defined as the difference in zonal mean temperature between the tropics (30S-30N) and the polar

---

[4]The WACCM4 simulations analyzed here are a lot more computationally expensive those in (Bittner et al., 2016), as they involve interactive ozone chemistry. In fact, we are aware of no other study with a coupled atmosphere-ocean-chemistry model which has analyzed ensembles with more than a handful of members. Just to cite a few recent studies: McLandress et al. (2011) analyze 3 members, Solomon et al. (2015) 6 members, Li et al. (2018) 4 members. So, we submit that a 13-member ensemble with interactive chemistry, and coupled ocean and sea-ice components, represents a susbtantial step forward.





cap (60-90N) at 50 hPa: that level is chosen so as to capture the maximum amplitude of the stratospheric warming from Mt. Pinatubo at low-latitudes. The relationship between the $U'_{10}$ and $\nabla T'_{50}$ is shown in Fig. 5a: their correlation is exceedingly high (with an $r^2$ value of 0.89). From the ensemble mean value (black dot) one can see that, indeed, a warming of the tropical lower stratosphere by a potent low-latitude eruption does indeed result in a stronger wintertime polar vortex in our model.

5    The key question, however, is: how much stronger? In the case of the Mt. Pinatubo eruption, this is given by the black circles in Fig. 5a, which indicate the ensemble mean value of 3.5 m/s for our WACCM4 simulation. This is in excellent agreement with the findings of Bittner et al. (2016), who also reported 1-2 m/s acceleration of the polar vortex following large low-latitude eruptions, and emphasized that 50-100 of ensemble members are need to establish this result in a statistically significant way. One cannot overemphasize how minuscule this forced response is when contrasted with the unforced, internal variability of the wintertime polar vortex, whose strength can vary by many tens of meters per second over a period as short as a week (e.g. during a stratospheric sudden warming event, which occur roughly every other year, see  Charlton and Polvani, 2007).

With this in mind, we now proceed to examining the second step of the proposed mechanism, the relationship between the polar vortex anomaly $U'_{10}$ and the Eurasian surface temperature anomaly $T'_s$. We find no meaningful correlation between the two, as evident from Fig. 5b (the $r^2$ value is 0.06). It is widely appreciated that the variability of the midlatitude tropospheric circulation is very large, so that it can easily overwhelm polar vortex anomalies of tens of meters per second. In fact, even stratospheric sudden warmings – which correspond to massive perturbations of the stratospheric polar vortex which results in a complete wind reversal, from westerlies to easterlies – are not always able to produce a significant surface signal (see the Sudden Warming Compendium, Butler et al., 2017).

Another way of illustrating the weakness of the connection between polar vortex strength and Eurasian surface temperature anomalies is to contrast two WACCM4 ensemble members – specifically #2 and #12 – for which $T'_s$ is shown in the top row of Fig. 6. We have chosen these two particular members as they simulate very similar Eurasian surface warming anomalies, not unlike the ones in the observations. In spite of those surface similarities, the corresponding stratospheric temperature gradients are completely different (see the middle row of Fig. 6). The tropical lower stratosphere is anomalously warm in both members, owing to the direct radiative effect of the volcanic aerosols, which is robust. In contrast the polar stratosphere is anomalously warm for one case (#2) but cold for the other (#12). The corresponding temperature gradients $\nabla T'_{50}$ are thus of opposite sign and, predictably, the polar vortex is anomalously weak for the former and strong for the latter member, as seen in the bottom row of Fig. 6, where we show the zonal mean zonal wind at 10 hPa. Note that these opposite-signed polar vortex anomalies have an amplitude of about 10 m/s, which is three times larger than the forced response documented above. In spite of such large and opposite-signed polar vortex anomalies, both members exhibit very similar surface temperature anomalies over Eurasia, as seen in the top row: this clearly demonstrates that polar vortex anomalies do not *necessarily* determine the surface anomalies.

For completeness, the full vertical structure of the ensemble mean temperature anomalies for the WACCM4 model is shown in Fig. 7a. The only statistically significant signal is found in the tropics, where WACCM4 greatly overestimates the post-Pinatubo warming, yielding a temperature gradient in the lower stratosphere that is considerably larger than the observed one: as seen in Fig. 5a, the ensemble-mean simulated value of $\nabla T'_{50}$ is 5.3°C , whereas the observed value is 0.4°C. In spite of a much larger temperature gradient anomaly than the observed one, we find little statistically significant response in the polar





stratospheric winds, as seen in Fig. 7b. There is an overall acceleration of the polar vortex, as one might expect, but the area of significance is quite small, and the grid point at 10 hPa and 60N (the canonical metric for the polar vortex strength) is not statistically significant.

This conclusion does not contradict the findings of Bittner et al. (2016), who reported a statistically significant response of the stratospheric polar vortex after the Mt. Pinatubo eruption in their model. We have only 13 members at our disposal here, and this is why we are unable to establish clear significance with WACCM4. To appreciate how difficult it is to obtain a statistically significant response in the polar vortex, we show the $U'_{10}$ anomalies for each of the 13 members in Fig. 8: there is a wide scatter across the 13 members, yielding an ensemble mean which is much smaller than most individual members. Nonetheless, the fact that only 4 members show a vortex weakening and the remaining 9 show a vortex strengthening is suggestive: it is quite likely that had we had 50 or 100 members in our ensemble, a statistically significant strengthening of the polar vortex would have emerged.

More important, however, is the red line in Fig. 8, showing the ERA-Interim anomalies: it indicates that the polar vortex was, actually, anomalously weak in the winter following the Mt. Pinatubo eruption. For clarity, we show the entire latitude/pressure profiles of the ERA-Interim temperature and wind anomalies in the bottom row of Fig. 7. Amazingly[5] enough, the polar stratosphere was anomalously warm (not cold) after the eruption (panel c), and the polar vortex was anomalous weak (not strong): note, in panel d, the negative zonal wind between 10 and 1 hPa, and between 50N and 60N, where the climatological polar vortex is located. So we conclude by asking: How can the stratospheric pathway mechanism be invoked as an explanation for the observed warming over the NH continents, if the polar stratosphere was actually *warmer* and the polar vortex was actually *weaker* in the winter that followed the 1991 eruption of Mt. Pinatubo?

## 5    Summary and Discussion

The aim of this paper has been to understand the cause of the warm anomalies that were observed over the NH continents following the Mt. Pinatubo eruption in June 1991. More specifically, referring back to the introduction, we have addresss two related but distinct questions: the ability of the models to simulate the observations and the importance of the stratospheric pathway.

First, we have clearly demonstrated that the current generation of coupled climate models is eminently capable of simulating such anomalies. Unlike previous studies, our conclusion follows from comparing the observed anomalies to *individual* model simulations, not to the *average* of multiple simulations. We have shown that climate models, when forced with an identical volcanic perturbation, can actually simulate a much larger warming than observed and, in fact, an equally large cooling. Furthermore, confirming many previous studies, we have shown that averaging across model simulations results in statistically insignificant surface temperature anomalies in the NH following the eruption. Taken together, and assuming climate models

---

[5]This crucial fact seems to have gone largely unnoticed in the literature. It is reported in the doctoral dissertation of Thomas (2008, see her Figures 4.16 and 4.17), and tangentially noted by Mitchell et al. (2011, see their Figure 8, and the accompanying text), who employed so-called "elliptical" diagnostics for the polar vortex. It is also briefly discussed in Toohey et al. (2014, see their Figure 1), who argue that wintertime stratospheric state in the first winter after Mt. Pinatubo may be not be representative of the "pure response" to the volcanic aerosols owing to confouding factors (e.g. the Quasi-Biennial Oscillation).





are not fundamentally flawed, these facts are here interpreted as follows: the internal variability of the climate system in the NH in wintertime is much larger than any impact from the Mt. Pinatubo eruption. As a consequence, it is hard to imagine that any substantial fraction of the observed warming anomalies in the NH during the 1991-1992 winter were caused by that volcanic eruption.

5    Second, we have examined in detail the potential role of an often invoked stratospheric pathway mechanism, which would allegedly mediate the signal from a low-latitude eruption to the higher-latitude continents by accelerating the polar vortex, and subsequently causing a positive phase of North Atlantic Oscillation (or the annular mode). Analyzing the WACCM4 model, which is a stratosphere-resolving model with interactive stratospheric ozone chemistry, we find the polar vortex acceleration accompanying the increased lower stratospheric temperature gradient after the Mt. Pinatubo eruption to be no larger than a few meters per second at best. And, we wish to emphasize, the WACCM4 model (like most others) produces an unrealistically large warming of the tropical lower stratosphere (see Figs.2d and 7a,b), which implies an unrealistically strong acceleration of the polar vortex. Even so, that acceleration is actually *not* statistically significant in our 13-member WACCM4 ensemble. This is in total agreement with the recent study of Bittner et al. (2016), who show that 50-100 members are needed to detect a significant acceleration of the polar vortex in the winter following a large-magnitude low-latitude eruption. This, in and of itself, is clear evidence that the forced polar vortex response is very small compared to the internal stratospheric variability in wintertime, where wind perturbations of many tens of meters per second are not unusual. And ultimately, in terms of affecting the tropospheric circulation and surface temperature, such small polar vortex anomalies are completely dwarfed by the internal tropospheric variability; this is why no statistically significant anomalies are found when averaging over many model simulations.

20    One might now ask how such evidence can be reconciled with several influential early studies, which have argued for the key role of the stratospheric pathway in causing the NH continental surface warming in the winter following the Mt. Pinatubo eruption. We suggest the following: those early models simply lacked a good representation of the stratosphere and, more crucially, of its variability, and this resulted in an overestimate of the forced response to the volcanic eruption. For instance, the model employed in Graf et al. (1993) had a mere 19 vertical levels in the vertical direction, with the model top at only 10 hPa. The same applies to the study of Kirchner et al. (1999), who improved the horizontal resolution but retained the same deficient vertical structure of their model. A severe lack of vertical resolution is also evident in the AMIP models analyzed in Mao and Robock (1998), all of which (with only one exception) have between 10 and 20 vertical levels (see Table 2 of Gates, 1992). Ditto for the study of Collins (2003): 19 vertical levels. As for Shindell et al. (2004), the two models used in that study have only 20 and 23 vertical levels, and the latter has a very coarse horizontal resolution as well ($8°$latitude $\times$ $10°$longitude): that model was, in fact, evaluated for its ability to simulate stratospheric sudden warmings, and found to greatly underestimate their frequency (see Fig. 3c of Charlton et al., 2007, under the item GISS23). The reader may want to contrast that model with the WACCM4 model used here, with 66 vertical levels, a model top at $\sim$140km, and an excellent simulation of the frequency of stratospheric sudden warmings (see Fig. 3a of Marsh et al., 2013).

A note is also in order regarding the recent study Zambri and Robock (2016). They reanalyzed a larger set of CMIP5 models than those in Driscoll et al. (2012), and considered only the anomalies in the first winter after the eruptions. From the multi-



model average anomalies following the two largest eruptions since the pre-industrial era they conclude that "the observed surface temperature anomalies are related to changes in the winter circulation *caused* by the volcanic eruptions" (emphasis added), a claim obviously at odds with much of the previous literature, and with the results presented here. However, as their conclusion was drawn by averaging anomalies from the 1883 Krakatau eruption with those from the 1991 Mt. Pinatubo erup-

tion, it is not immediately obvious how to disentangle the forced response to Mt. Pinatubo alone, which is the sole subject of the present study. We plan to carefully examine other volcanic eruptions in an upcoming paper.

Nonetheless, we have briefly analyzed other recent[6] eruptions simulated by the three models described in Section 2.1. Of particular interest is the 1982 eruption of El Chichón (Robock, 1983), which was also followed by anomalous wintertime warming over the Nothern Hemisphere continent (as shown in supplementary Fig. S6). As for the 1991 Mt. Pinatubo eruption,

all three models produce (1) a statistically insignificant forced response and (2) both warm and cold anomalies with identical volcanic forcing (see supplementary Fig. S7), indicating that the observed continental winter warming following the 1982 El Chichón eruption was also, very likely, a simple manifestation of internal variability. Of course, the validity of our interpretation is dependent on the models' ability to accurately simulated the internal variability of the climate system.

Still, leaving models – and their possible biases – aside, one could nonetheless argue that several studies have "demon-

strated", on the basis of various temperature reconstructions, that many low-latitude volcanic eruptions have been followed by NH continental warming in wintertime. Whether those demonstrations are truly convincing depends, crucially, on the quality of the surface temperature reconstructions and on the soundness of the methodology employed. Just to give an example: the early claim of Robock and Mao (1992) was based on the analysis of a single temperature dataset for – literally – one dozen eruptions, half of which occurred at latitudes outside 30S-30N, averaging together larger and smaller events, including a mixture of first

and second winter anomalies (depending on the eruptions). For the reasons stated in Section 2.2 above, we very much agree with the recent suggestion of Zambri and Robock (2016) that (1) only the first winter after each eruption should be considered, (2) eruptions of different magnitudes should not be averaged together : if these two procedural choices are important, many studies in the literature would need to be reconsidered.

In any case, going back to Mt. Pinatubo, the fact remains that from December 1991 to February 1992, the observed surface

temperatures were anomalously warm over North America and Eurasia, and that fact may deserve an explanation. Our analysis clearly shows that the continental warming that occurred in the first winter following the 1991 eruption was most likely a simple manifestation of internal atmospheric variability, and was completely unrelated to the eruption itself. So, the next question is: what might be the source of variability that resulted in the NH continental warming? An obvious candidate would be the El Niño-Southern Oscillation (ENSO) phenomenon, since it is well known that the eruption of Mt. Pinatubo corresponded with an

El Niño event (see, e.g., Lehner et al., 2016), which is believed to influence the North Atlantic and Eurasia in winter (Brönnimann, 2007; Rodríguez-Fonseca et al., 2016). Unfortunately, El Niño conditions are typically associated with a contraction of the tropical belt (Lu et al., 2008) and a negative phase of the North Atlantic Oscillation (Li and Lau, 2012), which is typically accompanied by cold anomalies over Eurasia. It is, therefore, difficult to argue that the observed post-Pinatubo continental

---

[6]After the 1963 eruption of Mt. Agung, the volcanic aerosol cloud spread primarily into the Southern Hemisphere (Viebrock and Flowers, 1968): that eruption is thus not the best candidate for exploring the causal link between low-latitude eruptions and anomalies over the Northern Hemisphere continents.





warming was caused by El Niño. In fact, there is some good modeling evidence confirming this. First, Thomas et al. (2009) reported a "very strong" response to El Niño in their model, that "can mask the effects due to volcanic warming". Second, analyzing so-called pacemaker[7] simulations with the CAM5-LE model, McGraw et al. (2016) show a large forced signal in the tropospheric circulation from El Niño in the Northern Hemisphere, which greatly resembles a negative annular mode (see their

5   Fig. 11f); they don't show surface temperatures, but one would easily expect cold anomalies over the NH continents in those simulations. If, then, El Niño needs to be ruled out, we may just have to admit that the intrinsic variability of the high latitude tropospheric circulation, which is known to be very large (Shepherd, 2014), might have to suffice as an explanation.

*Acknowledgements.* LMP and AB are grateful for the support of the US National Science Foundation (USNSF), and wish to express their gratitude to Dr. Ryan Neely for performing and making available some of the WACCM4 model integrations. Computing resources for

10  the WACCM4 and CAM5-LE were provided by the Computational and Information Systems Laboratory (CISL) at the National Center for Atmospheric Research (NCAR). We also acknowledge the National Oceanic and Atmospheric Administration's (NOAA) Research and Development High Performance Computing Program, for providing computing and storage resources for some of the WACCM4 runs, and wish to thank Henry LeRoy Miller for his assistance with NOAA's high performance computing facilities. The CESM Large Ensemble Project and supercomputing resources were provided by the USNSF and NCAR/CISL. We also acknowledge Environment and Climate

15  Change Canada's Canadian Centre for Climate Modelling and Analysis for executing and making available the CanESM2 Large Ensemble simulations used in this study, and the Canadian Sea Ice and Snow Evolution Network for proposing the simulations.

---

[7]In these simulations a fully coupled atmosphere-ocean is employed, but SST anomalies in the eastern tropical Pacific are nudged to observations, so as to faithfully simulate El Niño events.



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

**Figure 1.** Surface air temperature anomalies (in °C ) for the post-Pinatubo winter of 1991-92 relative to the reference period (1985-1990) in observations (a) GISTEMP and (b) HadCRUT4, and in reanalyses (c) NCEP/NCAR and (d) ERA-Interim.





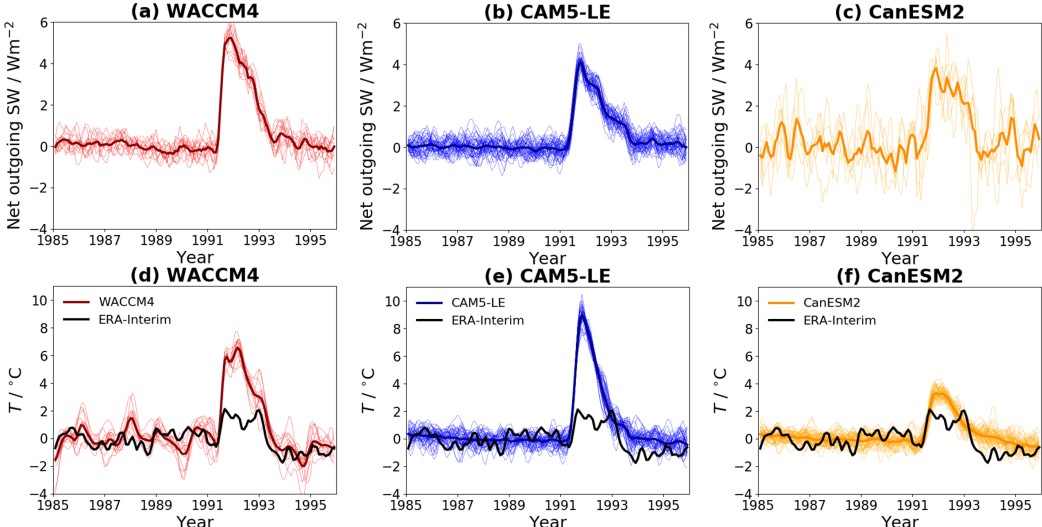

**Figure 2.** Top row: globally averaged, de-seasonalized, net, outgoing SW radiation at the top of the atmosphere (in W/m$^2$). Bottom row: tropically averaged (30S-30N), deseasonalized temperature (in °C ) at 50 hPa. Left column: WACCM4 (red lines). Middle column: CAM5-LE-LE (blue lines). Right column: CanESM2 (yellow lines). In each panel, the time series for each ensemble member (thin lines) and for the ensemble mean (bold line) are shown. In the bottom row, ERA-Interim values are also shown for comparison (black). All time series are anomalies from the 1985-1990 mean, and are smoothed with a 3-month running average, for direct comparison with Figs. 2 and 3 of Driscoll et al. (2012).




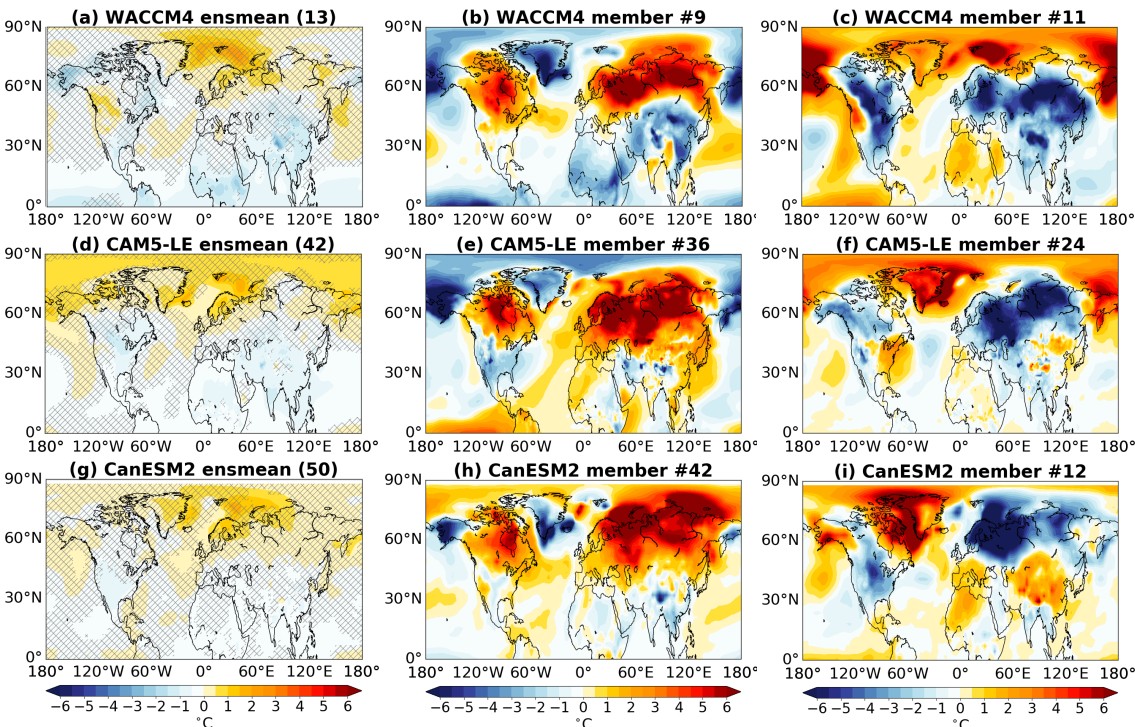

**Figure 3.** Wintertime surface air temperature anomalies (in °C ) as simulated by WACCM4 (top row), CAM5-LE (middle row) and CanESM2 (bottom row) following the 1991 Mt. Pinatubo eruption. Left column: the ensemble mean for each model (with the number of ensemble members in parentheses), and hatching over areas where the anomalies not significant at the 95% confidence level. Middle column: individual members exhibiting extreme warming over the NH continents for each model. Right column: individual members exhibiting extreme cooling.





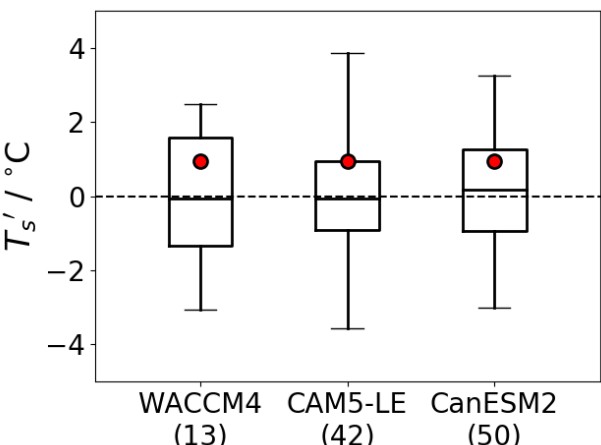

**Figure 4.** Box and whisker plots of simulated surface temperature anomaly (in °C ) over Eurasia (40-70N, 0-150W) in the first post-Pinatubo winter (1991-92) relative to the reference period (1985-1990). The horizontal line inside each box denotes the ensemble mean; the lower and upper limits of each nbox denote the 25th and 75th percentile values, respectively; the whiskers span the full range of the ensemble members. For comparison, the red circles denote the value calculated from the ERA-Interim reanalyses.



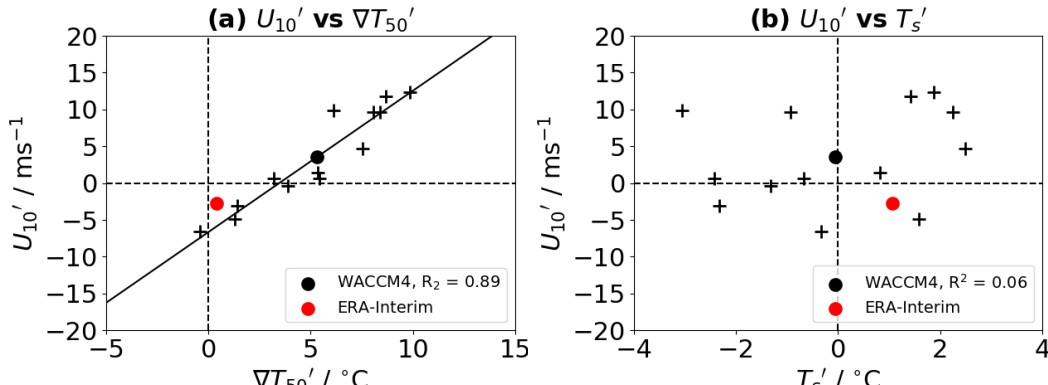

**Figure 5.** Scatter plots showing the relationship between $U'_{10}$, the anomalies in the zonal mean zonal wind at 10hPa and 60N (in m/s) and the anomalies in (a) the NH meridional temperature gradient $\nabla T'_{50}$ between the tropics (30S-30N) and the pole (60-90N) (in °C ), and (b) the Eurasian surface air temperature $T'_s$ (also in °C ). Crosses show individual ensemble members, and the black dot shows the ensemble mean value. The red dot shows the ERA-Interim reanalysis.



**Figure 6.** The surface temperature $T_s'$ (top), the zonal mean temperature $T'$ (middle) and 10h hPa zonal mean zonal wind $U_{10}'$ anomalies for WACCM4 member #2 (left) and member #12 (right)





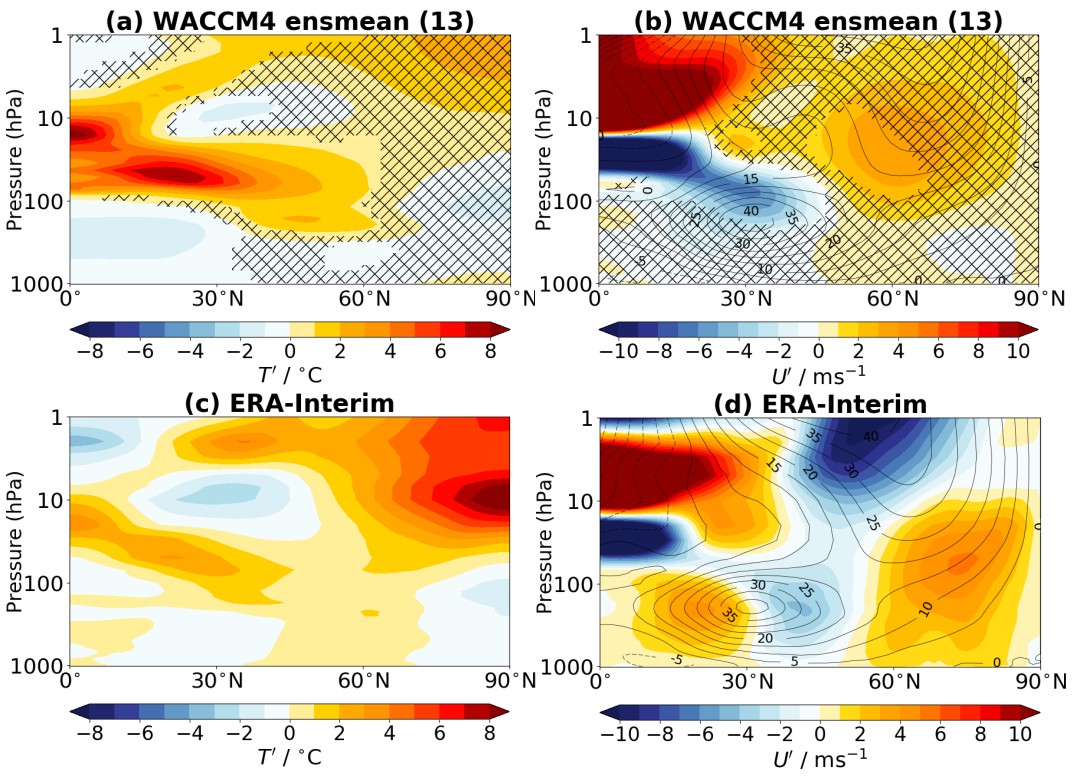

**Figure 7.** Latitude/pressure anomalies for the winter following the 1991 Mt. Pinatubo eruption. Left: zonal mean temperature ($T'$). Right: zonal mean zonal wind $U'$, with the climatology in black contours. Top: the ensemble mean of the WACCM simulations, with hatching for values that are not significant at the 95% confidence level. Bottom: corresponding anomalies in the ERA-Interim reanalysis.



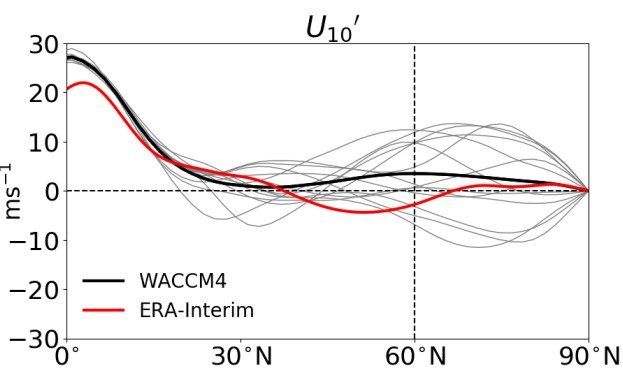

**Figure 8.** Zonal mean zonal wind anomalies at 10 hPa ($U'_{10}$) vs. latitude, for the individual WACCM4 simulations (gray), for the ensemble mean (black), and for the ERA-Interim reanalysis (red).