# Peer review of "Northern Hemisphere continental winter warming following the 1991 Mt. Pinatubo eruption: Reconciling models and observations"

_Atmospheric Chemistry and Physics, 2018_

## Referee Comment (RC1) · Anonymous Referee #2 · 11 Jun 2018

This manuscript argues that Mt. Pinatubo's 1991 eruption had little to no impact on continental surface temperatures, and hence observed surface warming in midlatitude winters was due to natural variability. The implication is that similar conclusions may hold for other eruptions. Indeed, there has been perhaps excessive focus on explaining and simulating the observed Pinatubo response, without sufficient regard to the inherent role of natural variability.

This manuscript contributes to ongoing discussion by frankly addressing the issue of natural variability, and its novel employment of a coupled atmosphere-ocean-chemistry model for volcanic simulation is a useful addition to the literature, even if having only

13 members for that model leads to difficulties with statistical significance.

After tempering the overall claims and investigating a lower-stratospheric pathway as detailed below, this manuscript is suitable for publication.

**General comments**

1. The text is too quick to dismiss temperature reconstructions. Despite the inherent uncertainties of temperature reconstruction, the key is that averaging over several centuries reveals a statistically significant pattern of winter warming, apparently even stronger for the subsequent winter (Fischer et al., Geophys. Res. Lett. 2007), which would be highly coincidental if volcanic eruptions were unrelated.

2. P5 L19 and elsewhere describes the ensemble sizes as "large" (13, 42, and 50 members). However, P3 L12 mentions that Bittner et al. (2016) needed 60 members for 95% confidence in the stratospheric response, and other comparable examples are mentioned. Thus "large ensemble" seems incorrect a priori, so the difficulties throughout in achieving significance should not be surprising.

3. To address the underlying mechanism(s), the manuscript uses a 13-member ensemble with an improved stratosphere to argue against a pathway between stratospheric vortex perturbations and surface temperature perturbations. However, the discussion does not address the possibility of a lower-stratospheric pathway, which from Fig. 7 seems plausible. This could be important if the impact on the vortex does not have the same vertical structure as natural variability. Addressing this would be straightforward by repeating the analysis of Fig. 5 at one or two lower levels such as 50 hPa.

**Specific comments**

P1 L17: "is likely to be very small" is unclear—it was likely very small based on these model results?

P2 L1: "short lived" is relative; P6 L12 cites an e-folding time of 12 months, longer than most natural modes of variability.

P2 L16–25: The suggestive tone is biased. In the "widely believed" (P1 L1) viewpoint of winter warming, it should not be "remarkable" that the subsequent winter after Pinatubo "happened to be" warm, nor is it "highly perplexing" and "difficult to reconcile" that historical warming is not exactly correlated with eruption magnitude. Rather, the question is whether eruptions of a given strength can induce a statistically and physically significant winter warming. Missing here is Fischer et al. (Geophys. Res. Lett. 2007), which should be cited here as the most recent (known to this reviewer) post-eruption temperature reconstruction.

P3 L5: An implicit assumption of this mechanism is that the balanced acceleration lies in the vortex region, which is not necessarily the case. Bittner et al. (J. Geophys. Res. Atmos. 2016) discusses this.

P3 L13: "tiny" is relative; perhaps relate this to annular mode, standard deviation of lowpass-filtered winds, or similar. See also comment for P9 L13–15.

P3 L19: Stenchikov et al. (2002) had only 4 ensemble members, and argued for a reduction in planetary wave activity, contrary to what Graf et al. (2007) said about a single eruption. Perhaps this paragraph should conclude that the mechanism is not demonstrated by these single-model, small-ensemble studies.

Remove "clearly, "abundantly clear," etc. for P3 L21, P7 L24, P8 L5, P9 L34, P10 L11, P10 L29, P11 19, P12 L22. The word doesn't enhance the argument and may come across as proof by intimidation.

P4 L11–14: should also cite Barnes et al. (J. Clim. 2016), which does find a significant response. Thus even when experimental design and intermodel spread are controlled

for, the result can still vary!

P5 L30–P6 L7: a common limitation, here and in many other studies, is that it is unknown whether or not the response is linear. (Perhaps a threshold magnitude of forcing is necessary, or stronger forcing induces feedbacks.) This limitation should be stated, as the conclusions for these Pinatubo-sized eruptions may not hold for smaller or larger eruptions.

P7 L23: ensemble averaging reduces, but does not eliminate, internal variability.

P7 L25: in F3, the larger ensembles have slight windows of significance. An estimate (even via simple bootstrapping) of the necessary number of ensemble members to achieve continental-scale significance would be very helpful for this discussion and for future studies.

P7 L35: internal variability is not superimposed to any forced response—it may very well be nonlinear (i.e., the higher moments of the underlying probability distribution functions may change).

P7 L11 and F4: rather than a box-and-whisker plot, a plot of the 3 probability distribution functions is preferable here in my opinion, so that the reader can compare the distributions.

P8 L25–26: the other two models may not have as accurate a representation of the stratosphere, but do they give similar results? If so, they should be included. If not, why is the subsequent comparison with Bittner et al. (2016) (which similarly has a non-interactive stratosphere) valid but not with the other two?

P9 L13–15: it is not appropriate to compare weekly SSW variability with volcanic forcing as their timescales are well-separated. The appropriate comparison would be something like variability of DJF average, which is approximately 10 m/s at 10 hPa and 6 m/s at 50 hPa, more comparable to the 3.5 m/s reported here.

P9 L16–17 and F5: It might be helpful to add a third scatter plot of a lower comparison

point like $\nabla T_{50}$ and $T_s$, which may correlate better than 10 hPa, if the perturbation is comparatively larger than natural variability.

P9 L23–34: examining two individual ensemble members does not offer any insight into the mechanism, especially since the manuscript already argues that natural variability is large. This paragraph and the corresponding F6 should be removed.

P10 L7: even if 10 hPa is "canonical," it may be the wrong level for finding the mechanism. Repeating the same analysis at a lower level, such as 50 hPa, would either strengthen the current null-hypothesis argument or provide new insight into the mechanism's vertical extent.

P10 L14–16: "quite likely" and "would have emerged" are purely speculative and should be removed, as the null hypothesis was not rejected by the significance test. Instead, a simple bootstrapping estimate of the requisite number of samples to achieve significance may again be helpful.

P10 L17–24: again, the possibility of a lower stratospheric pathway should, and could easily, be addressed here with the existing methodology.

P11 L28–30: are their low model tops thus an indirect argument for a lower stratospheric pathway? P10 L25 to P13 L4: the conclusions should be updated following any relevant changes made as a result of these comments.

**Technical comments**

P8 L6: "stonger" should be "stronger"

P8 L11: "Fig. 5" should be "Fig. 4"

P9 L2: "need" should be "needed"

F4: "nbox" should be "box"

F5: "$R_2$" in legend of subplot (a) should be "$R^2$"

---

## Referee Comment (RC2) · Anonymous Referee #3 · 20 Jun 2018

The paper deals with a longstanding issue of the inability of climate models to reproduce the high-latitude near-surface winter warming following the major low-latitude volcanic eruptions. I appreciate the authors have risen this issue again using a set of the "new generation" models. I believe the reviving this issue is useful but I cannot completely agree with some interpretations and methodology the authors use in this study.

General comments:

To test the mechanism based on the troposphere-stratosphere dynamic interaction, the authors conducted the Pinatubo case study focusing on the first winter after the June

1991 volcanic explosion in the Philippines. However, the choice of the case-study is unfortunate as in the winter of 1991/92 the positive AO was not forced by the "stratospheric" mechanism. In observations, the polar vortex was weak and asymmetric with the wave number 2 prevailing. So, it is pointless to analyze this response to prove or disprove the stratosphere/troposphere dynamic interaction mechanism. Stenchikov et al. (2004) indicated that the easterly QBO phase in winter of 1991/92 weakened the polar vortex, and winter of 1992/93 with a westerly QBO phase provides a better case-study to test the "stratospheric" mechanism.

As it is correctly stated in P8, L22-24, the "stratospheric" mechanism involves two steps: strengthening of the stratospheric polar vortex and downward propagation of the signal. The proof of the latter portion of the mechanism did not come directly from "volcanic" studies, as volcanic eruptions are rear and provide insufficient statistics, but from climatological studies of Baldwin and Dunkerton (1999). As mentioned by Stenchikov et al. (2006) the strengthening of the polar vortex caused by the equatorial lower stratospheric warming due to aerosol-induced heating, is robust in the models, but the models fail to reproduce the downward transport. So, to disprove this "stratospheric" mechanism the authors have to deal with the climatological analysis as well.

It is not surprising that some of the model ensemble members could produce a "winter warming" pattern. It is more important how frequently this pattern appears and what mechanism causes it. Models have to produce this pattern more frequently to be consistent with the climatological studies that show a statistically significant positive AO pattern after compositing multiple equatorial eruptions. The conclusion that the up-to-date models could perfectly reproduce the winter warming based on the fact that some ensemble members capture it, is not supported.

Specific comments:

P3, L13-15: A vertical propagation of the planetary waves is a threshold process as suggested by Charney and Drazin (1961), so small change of the wind could qualitatively change the planetary wave reflection coefficient.

P4, L26-27: AO response is an atmospheric effect. Why increasing of model complexity should matter to answer the question that the Stratosphere-Troposphere Interaction is real.? E.g., if ozone additional radiative effect matters, this has to be specifically shown.

P5, L23: The chosen models are inconsistent in reproducing the aerosol forcing. In Figure 2 the aerosol forcing in the models differs by 50%. It would be useful to mention what was the observed forcing to compare with.

P5, L30-33: The winter of 1991/92 after the Pinatubo eruption is a wrong choice (see Figure 5). A "composite" approach has to be considered to obtain statistically significant anomalies in observations.

P6, L2: This is incorrect. The eruption of Mt Agung of 1963 developed an aerosol equatorial reservoir that caused warming of the equatorial lower stratosphere and enhanced equator-pole temperature gradient in the lower stratosphere. The re-distribution of aerosols between the hemispheres is not directly relevant.

P6, L8: The first winter is a wrong choice.

P6, L12: Volcanic aerosols remain in the equatorial reservoir in the second winter after the eruption that is why the effect is seen in the second winter as well.

P6, L18: Driscoll et all. (2012) adopted this methodology from Stenchikov et al. (2006).

P7, L8-9: The shortwave (SW) radiative forcing in three chosen models differs by 50%. There is much more differences in SW and Longwave (LW) aerosol absorption.

P7, L13-15: The models three times overestimate the equatorial lower stratospheric heating caused by volcanic aerosols. This is the main forcing of the stratosphere-troposphere dynamic interaction. There is something wrong here.

P7, L20: "With this IN mind"

**[ACPD](ACPD)**

Interactive
comment

P7, L32-35: I think the correct question to ask is whether models are able to correctly reproduce the probability distributions of the Arctic oscillation (AO) responses to volcanic forcing. But for this one has to extract multiple cases from observations and to construct the observed probability distribution. It is not doable with only one post-eruption season considered.

P9, L10: Planetary wave reflection is the threshold process (Charney and Drazin, 1961) when small changes matter.

P9, L14: The wind variability coming from SSW is not relevant to the process. As soon as polar vortex zonal wind weakens below the threshold, planetary waves can propagate upward nonlinearly weakening the polar vortex. So, the amplitude of wind changes below the threshold, no matter how large it is, does not count. Sampling has to focus on the strong vortex cases for this purpose.

P10, L17-18: Exactly, the winter of 1991/92 is not suitable to study the forced stratosphere-troposphere dynamic interaction, as the positive phase of AO in the troposphere was caused by a different mechanism.

P11, L2: You mean surface cooling/warming, not in the lower stratosphere. Please clarify.

P11, L5-8: You have to explain why do we see a positive AO anomaly climatologically after multiple volcanic eruptions. If this would be extremely rear events as in the models, then a positive AO anomaly would not be seen in observations.

P11, L15-23: The strengthening of the polar vortex caused by the volcanic aerosols heating in the lower equatorial stratosphere is robust. This is the threshold process, so a weak strengthening matters. And it is unfair to apply wind variability in SSW to scale the increase in maximum wind.

P11, L25-35: The downward propagation mechanism was proved using climatological analysis (Baldwin and Dunkerton, 1999) and has to be challenged on this basis.
P12, L25-30: ENSO definitely could affect surface temperatures, although Volcano-ENSO interaction is highly nonlinear. At least contribution of ENSO variability in the volcanic signal has to be removed properly, which was never done in this analysis. Another important mode of variability is QBO that was not considered and reported in this study. QBO plays an important role in stratospheric wave propagation and could directly affect polar vortex and shape the stratosphere-troposphere dynamic interaction.

---

## Author Comment (AC1) · 17 Dec 2018

The authors' replies to the referees comments can be found *in italic* below each comment.
* * *
**REPLIES TO REFEREE # 2:**

This manuscript argues that Mt. Pinatubo's 1991 eruption had little to no impact on continental surface temperatures, and hence observed surface warming in midlatitude winters was due to natural variability. The implication is that similar conclusions may hold for other eruptions. Indeed, there has been perhaps excessive focus on explaining and simulating the observed Pinatubo response, without sufficient regard to the inherent role of natural variability.

This manuscript contributes to ongoing discussion by frankly addressing the issue of natural variability, and its novel employment of a coupled atmosphere-ocean-chemistry model for volcanic simulation is a useful addition to the literature, even if having only 13 members for that model leads to difficulties with statistical significance. After tempering the overall claims and investigating a lower-stratospheric pathway as detailed below, this manuscript is suitable for publication.

**1. General comments:**

1. The text is too quick to dismiss temperature reconstructions. Despite the inherent uncertainties of temperature reconstruction, the key is that averaging over several centuries reveals a statistically significant pattern of winter warming, apparently even stronger for the subsequent winter (Fischer et al., Geophys. Res. Lett. 2007), which would be highly coincidental if volcanic eruptions were unrelated.

*Thanks for the suggestion: we now cite Fischer et al (2007), with the appropriate caveats.*

2. P5 L19 and elsewhere describes the ensemble sizes as "large" (13, 42, and 50 members). However, P3 L12 mentions that Bittner et al. (2016) needed 60 members for 95% confidence in the stratospheric response, and other comparable examples are mentioned. Thus "large ensemble" seems incorrect a priori, so the difficulties throughout in achieving significance should not be surprising.

*The term "large ensemble" is widely used in the literature to refer to the both the CAM5-LE and CanESM2 datasets. We noted in the manuscript that our relatively modest 13-member WACCM ensemble might not be best described as "large": however, it is the largest ensemble of chemistry-climate model integrations studied to date. This is why we use the term here.*

*As for the lack of a surface temperature response: we find none in our 13-member WACCM ensemble, and neither did Bittner et al (2016) in their larger 100-member ensemble (see figure 6.4 of this thesis). And neither did Driscoll et al (2012) using many CMIP5 model runs. So the "difficulties in achieving significance" are very robust result across the literature, and have little to do with the relatively small size of our WACCM ensemble.*

3. To address the underlying mechanism(s), the manuscript uses a 13-member ensemble with an improved stratosphere to argue against a pathway between stratospheric vortex perturbations and surface temperature perturbations. However, the discussion does not address the possibility of a lower-stratospheric pathway, which from Fig. 7 seems plausible. This could be important if the impact on the vortex does not have the same vertical structure as natural variability. Addressing this would be straightforward by repeating the analysis of Fig. 5 at lower levels such as 50 hPa.

*Following the referee's suggestion, we have recomputed Fig. 5 using U at 50 hPa (see below). It is very similar to the one using U at 10 hPa, which we included in the paper. The mean surface temperature response at the surface is zero (the black dot in the right panel), with 7 members showing cooling and 6 members showing warming.*

[Figure]

*Furthermore, we combine the two scatter plots into one, to directly show that there is no connection between the temperature gradient at 50 hPa (which is affected by volcanic aerosols) and the Eurasian surface temperature (which is not). This is shown below.*

[Figure]

*We hope this will suffice to convince the referee that the wind and temperature-gradient anomalies in the stratosphere (whether upper or lower) are not the cause of the surface temperature anomalies. This is the key point of our paper: there is no mechanism to be explained. Internal atmospheric variability suffices to produce the surface anomalies.*

**2. Specific comments:**

P1 L17: "is likely to be very small" is unclear – it was likely very small based on these model results?

> *Thank you for the suggestion. We have added "in our models" to clarify the sentence.*

P2 L1: "short lived" is relative; P6 L12 cites an e-folding time of 12 months, longer than most natural modes of variability.

> *We are using the expression "short lived" as it refers to a forcing of the climate system. The volcanic forcing is short-lived compared to anthropogenic forcings (e.g. the multi-decadal increase in carbon dioxide) or the solar forcing (whether the 11-year solar cycle or longer fluctuations of the solar constant).*

P2 L16–25: The suggestive tone is biased. In the "widely believed" (P1 L1) viewpoint of winter warming, it should not be "remarkable" that the subsequent winter after Pinatubo "happened to be" warm, nor is it "highly perplexing" and "difficult to reconcile" that historical warming is not exactly correlated with eruption magnitude. Rather, the question is whether eruptions of a given strength can induce a statistically and physically significant winter warming. Missing here is Fischer et al. (Geophys. Res. Lett. 2007), which should be cited here as the most recent (known to this reviewer) post-eruption temperature reconstruction.

> *As mentioned above, we have now cited the Fischer et al (2007) paper, with the appropriate caveats. As for the biased tone: we are simply saying that one expects surface cooling from a strong volcanic eruption, so any surface warming is surprising to us. Perhaps we are too naive, but naivete is not bias: we are simply offering our viewpoint.*

P3 L5: An implicit assumption of this mechanism is that the balanced acceleration lies in the vortex region, which is not necessarily the case. Bittner et al. (J. Geophys. Res. Atmos. 2016) discusses this.

> *That may be the case, but we are simply summarizing the "widely believed" viewpoint as stated, e.g. by Robock (Science, 2002), where one can read: The polar vortex is strengthened by lower stratosphere warming at low latitudes, which is caused by absorption of solar and terrestrial radiation by the volcanic aerosol cloud.*

P3 L13: "tiny" is relative; perhaps relate this to annular mode, standard deviation of lowpass-filtered winds, or similar. See also comment for P9 L13–15.

> *We have rephrased the sentence.*

P3 L19: Stenchikov et al. (2002) had only 4 ensemble members, and argued for a reduction in planetary wave activity, contrary to what Graf et al. (2007) said about a single eruption. Perhaps this paragraph should conclude that the mechanism is not demonstrated by these single-model, small-ensemble studies. Remove "clearly", "abundantly clear," etc. for P3 L21, P7 L24, P8 L5, P9 L34, P10 L11, P10 L29, P11 19, P12 L22. The word doesn't enhance the argument and may come across as proof by intimidation.

*We have removed most instances of the word "clearly" where suggested by the reviewer.*

P4 L11–14: should also cite Barnes et al. (J. Clim. 2016), which does find a significant response. Thus even when experimental design and intermodel spread are controlled for, the result can still vary!

*Thank for the suggestion. However, Barnes et al. (J. Clim. 2016) find that the CMIP5 model response of the circulation in the NH projects very poorly on the annular mode (which contradicts with the originally proposed mechanism); also they do not explicitly examine Eurasian temperatures in wintertime.*

P5 L30–P6 L7: a common limitation, here and in many other studies, is that it is unknown whether or not the response is linear. (Perhaps a threshold magnitude of forcing is necessary, or stronger forcing induces feedbacks.) This limitation should be stated, as the conclusions for these Pinatubo-sized eruptions may not hold for smaller or larger eruptions.

*We agree with the reviewer. We have added a sentence to that effect.*

P7 L23: ensemble averaging reduces, but does not eliminate, internal variability.

*Thanks for noting this. We have corrected the sentence.*

P7 L25: in F3, the larger ensembles have slight windows of significance. An estimate (even via simple bootstrapping) of the necessary number of ensemble members to achieve continental-scale significance would be very helpful for this discussion and for future studies.

*The areas of significance are non existent for WACCM and minuscule for the two low-top models. Also, recall that Bittner (2015) reported no significant warming over Eurasia, even with a 100 member ensemble. If hundreds of members are needed to produce a significant warming, the suggested bootstrapping calculation is an academic exercise.*

P7 L35: internal variability is not superimposed to any forced response – it may very well be non-linear (i.e., the higher moments of the underlying probability distribution functions may change).

*Following Deser et al (2012), we decompose the anomalies in any one realization as a sum of a forced response (defined as the ensemble mean anomaly) and the internal variability (the difference). Hence our use of the word "superimposed". This procedure is standard, we think, in all papers that have analyzed large ensembles of model simulations.*

P7 L11 and F4: rather than a box-and-whisker plot, a plot of the 3 probability distribution functions is preferable here in my opinion, so that the reader can compare the distributions.

*The whisker plots are PDFs. They show the mean, the percentile ranges and the full extent of each ensemble. Also, in Fig 4 the y-axis is identical: the reader can immediately and quantitatively compare these three distributions.*

P8 L25–26: the other two models may not have as accurate a representation of the stratosphere, but do they give similar results? If so, they should be included. If not, why is the subsequent comparison with Bittner et al. (2016) (which similarly has a non-interactive stratosphere) valid but not with the other two?

*The CAM-LE and CanESM are "low-top models": as such they do not simulate the observed stratospheric variability, e.g. Stratospheric Sudden Warming events. They are, therefore, inappropriate for examining the stratospheric pathway. This is why we focus our discussion on the WACCM ensemble in the paper.*

*Nonetheless, following the referee's suggestion, we now have added to the supplementary material the equivalent of Fig. 5 for the other two modes. The results are similar, and the very fact these low top models give similar results to WACCM is, of itself, a demonstration that the stratospheric pathway is not needed to explain the surface warming anomalies.*

P9 L13–15: it is not appropriate to compare weekly SSW variability with volcanic forcing as their timescales are well-separated. The appropriate comparison would be some- thing like variability of DJF average, which is approximately 10 m/s at 10 hPa and 6 m/s at 50 hPa, more comparable to the 3.5 m/s reported here.

*We politely disagree. The point we are making is that even SSWs, which are huge disruptions of the stratospheric circulation, are often incapable of reaching the surface. Hence, it is difficult to imagine how a 1-2 m/s wind anomaly from Mt. Pinatubo would result in strong Eurasian warming. That amplitude is too small, and the accompanying surface signal is swamped by the internal variability.*

P9 L16–17 and F5: It might be helpful to add a third scatter plot of a lower comparison point like $\nabla T_{50}$ and $T_s$, which may correlate better than 10 hPa, if the perturbation is comparatively larger than natural variability.

*This suggestion has been addressed above, in the "General Comments" section.*

P9 L23–34: examining two individual ensemble members does not offer any insight into the mechanism, especially since the manuscript already argues that natural variability is large. This paragraph and the corresponding F6 should be removed.

*We politely disagree. There is much recent literature on understanding internal climate variability using large ensembles, and showing two individual members of the ensemble is the simplest and most immediate way to visually convey the importance of internal variability, which often overwhelms the forced response. This was done, for instance, in the papers below, just to cite a few examples.*

- *Deser, C., R. Knutti, S. Solomon, and A. S. Phillips: Communication of the role of natural variability in future North American climate. Nat. Clim. Change (2012)*
- *Deser, C., et al.: Projecting North American Climate over the next 50 years: Uncertainty due to internal variability. J. Climate (2014)*
- *Deser, C., J. W. Hurrell and A. S. Phillips: The Role of the North Atlantic Oscillation in European Climate Projections. Clim. Dyn. (2017)*

P10 L7: even if 10 hPa is "canonical," it may be the wrong level for finding the mechanism. Repeating the same analysis at a lower level, such as 50 hPa, would either strengthen the current null-hypothesis argument or provide new insight into the mechanism's vertical extent.

*This suggestion has been addressed above, in the "General Comments" section.*

P10 L14–16: "quite likely" and "would have emerged" are purely speculative and should be removed, as the null hypothesis was not rejected by the significance test. Instead, a simple bootstrapping estimate of the requisite number of samples to achieve significance may again be helpful.

*Thanks. Bittner et al (2016) showed that a 100-member ensemble yields a significant vortex response. This is what we are referring to. We have rephrased that sentence.*

P10 L17–24: again, the possibility of a lower stratospheric pathway should, and could easily, be addressed here with the existing methodology.

*This suggestion has been addressed above, in the "General Comments" section.*

P11 L28–30: are their low model tops thus an indirect argument for a lower stratospheric pathway?

> *To the contrary! Poor vertical resolution in the stratosphere and a low model top suppress stratospheric variability, giving the false impression that the forced response is dominant. As models have added more levels and raised the top over the years, the forced surface warming has disappeared (in the CMPI3 and CMIP5 models).*

P10 L25 to P13 L4: the conclusions should be updated following any relevant changes made as a result of these comments.

> *There have been no relevant changes we are aware of.*

**Technical comments:**

P8 L6: "stonger" should be "stronger"

> *Fixed. Thank you.*

P8 L11: "Fig. 5" should be "Fig. 4"

> *Fixed. Thank you.*

P9 L12: "need" should be "needed"

> *Fixed. Thank you.*

F4: "nbox" should be "box"

> *Fixed. Thank you.*

F5: "$R_2$" in legend of subplot (a) should be "$R^2$"

> *Fixed. Thank you.*

**REPLIES TO REFEREE # 3:**

The paper deals with a longstanding issue of the inability of climate models to reproduce the high-latitude near-surface winter warming following the major low-latitude volcanic eruptions. I appreciate the authors have risen this issue again using a set of the "new generation" models. I believe the reviving this issue is useful but I cannot completely agree with some interpretations and methodology the authors use in this study.

**1. General comments:**

To test the mechanism based on the troposphere-stratosphere dynamic interaction, the authors conducted the Pinatubo case study focusing on the first winter after the June 1991 volcanic explosion in the Philippines. However, the choice of the case-study is unfortunate as in the winter of 1991/92 the positive AO was not forced by the "stratospheric" mechanism. In observations, the polar vortex was weak and asymmetric with the wave number 2 prevailing. So, it is pointless to analyze this response to prove or disprove the stratosphere/troposphere dynamic interaction mechanism. Stenchikov et al. (2004) indicated that the easterly QBO phase in winter of 1991/92 weakened the polar vortex, and winter of 1992/93 with a westerly QBO phase provides a better case-study to test the "stratospheric" mechanism.

> *As the title of the paper makes clear, our goal is to understand what happened over the NH continents in the winter following the Pinatubo eruption, and to reconcile models and observations. Pinatubo is the largest, most recent, best observed, low-latitude eruption: as such it needs to be understood before any other, much older and poorly observed eruptions. In fact, it is routinely used as the "poster child" for the impact volcanic aerosols on NH continental temperatures, e.g. Robock (Science, 2002).*

> *The prevailing narrative has been that the "models are missing something" as they show no surface warming after averaging many runs (of the same or of different models). Our paper argues that this an erroneous interpretation. The model average shows no warming because there is no significant warming response. It's that simple.*

> *The referee suggests that Pinatubo is a bad choice to test the "stratospheric mechanism" because the positive AO was not forced by that mechanism after that eruption. We agree: is was not not forced by the "stratospheric" mechanism and, in fact, it was not forced by any other mechanism. It was not forced at all. It was just variability. This is what our analysis of the three large ensembles very clearly demonstrates.*

> *As for the possible role of the QBO. First, recent studies (Bittner et al 2016, Robock and Zambri 2016) are agreed that only the first winter after the eruption should be averaged: that is when the aerosol presence is largest. Second, the simple fact that the QBO phase can wipe out the volcanic signal confirms our claim: that internal variability (of which the QBO is one aspect) overwhelms any forced response (should one exist). With large ensembles we can actually quantify the forced response, and we find that is is small and confined to the stratosphere. This is what Bittner et al (2016) also found.*

As it is correctly stated in P8, L22-24, the "stratospheric" mechanism involves two steps: strengthening of the stratospheric polar vortex and downward propagation of the signal. The proof of the latter portion of the mechanism did not come directly from "volcanic" studies, as volcanic eruptions are rear and provide insufficient statistics, but from climatological studies of Baldwin and Dunkerton (1999). As mentioned by Stenchikov et al. (2006) the strengthening of the polar vortex caused by the equatorial lower stratospheric warming due to aerosol-induced heating, is robust in the models, but the models fail to reproduce the downward transport. So, to disprove this "stratospheric" mechanism the authors have to deal with the climatological analysis as well.

> *The strengthening of the polar vortex was **not** observed the first winter after the Pinatubo eruption, and yet surface warming **was** observed. Therefore that surface warming could not have been caused by a stronger polar vortex (see Fig. 8).*

It is not surprising that some of the model ensemble members could produce a "winter warming" pattern. It is more important how frequently this pattern appears and what mechanism causes it. Models have to produce this pattern more frequently to be consistent with the climatological studies that show a statistically significant positive AO pattern after compositing multiple equatorial eruptions. The conclusion that the up-to-date models could perfectly reproduce the winter warming based on the fact that some ensemble members capture it, is not supported.

> *The main finding of our paper is that the models do not produce the winter warming more frequently after the Pinatubo eruption, because the ensemble average is zero. Warming happens half of the time, as a simple consequence of internal variability.*
>
> *We did not conclude that "models could perfectly reproduce the winter warming based on the fact that some ensemble members capture it". Our key finding, after analyzing three large ensembles, is that the observed warming falls well within the distribution of the model members. From this we conclude that the models capture the observations.*

**2. Specific comments:**

P3, L13-15: A vertical propagation of the planetary waves is a threshold process as suggested by Charney and Drazin (1961), so small change of the wind could qualitatively change the planetary wave reflection coefficient.

> *This is a linear result from highly idealized theoretical studies. Whether and how it may be relevant in practice remains to be demonstrated.*

P4, L26-27: AO response is an atmospheric effect. Why increasing of model complexity should matter to answer the question that the Stratosphere-Troposphere Interaction is real? E.g., if ozone additional radiative effect matters, this has to be specifically shown.

> *The literature on the impact of stratospheric ozone the annular mode is quite large. The referee might wish to consult, e.g. Thompson et al, (Nature Geoscience, 2011).*

P5, L23: The chosen models are inconsistent in reproducing the aerosol forcing. In Figure 2 the aerosol forcing in the models differs by 50%. It would be useful to mention what was the observed forcing to compare with.

*There is nothing peculiar about the models we have analyzed. They are of the same kind as those analyzed in Driscoll et al (2012) or Robock & Zambri (2016). In Fig. 2 we show the ERA-Interim temperature time series: one can see that WACCM and CAM5-LE simulate excessive warming, a common bias, as noted by Driscoll et al (2012).*

P5, L30-33: The winter of 1991/92 after the Pinatubo eruption is a wrong choice (see Figure 5). A "composite" approach has to be considered to obtain statistically significant anomalies in observations.

*See our reply to this in the General Comments section above as to why Pinatubo is not the wrong choice. As for adopting a "composite" approach: we have done so, by averaging all members of each ensemble. What we find, in agreement with Driscoll et al (2012) and Bittner (2015), that the surface response is not statistically significant. There is no reason to expect, a priori, that the surface anomaly should be significant, unless one thinks that internal variability is small, which is not the case (see Fig 3).*

P6, L2: This is incorrect. The eruption of Mt Agung of 1963 developed an aerosol equatorial reservoir that caused warming of the equatorial lower stratosphere and enhanced equator-pole temperature gradient in the lower stratosphere. The re-distribution of aerosols between the hemispheres is not directly relevant.

*We politely disagree: the hemispheric distribution is likely to matter. In any event, we have not analyzed the Mt Agung eruption, so the whole point in nugatory.*

P6, L8: The first winter is a wrong choice.

*We politely disagree: the aerosols forcing in the stratosphere is largest in the first winter.*

P6, L12: Volcanic aerosols remain in the equatorial reservoir in the second winter after the eruption that is why the effect is seen in the second winter as well.

*We politely disagree. As already pointed out, others have also concluded that only the first winter should be analyzed (Bittner et al 2019, Robock and Zambri 2016). The amount of volcanic aerosols in the winter 1992-93 was only a small fraction the one in 1991-92: see, for instance, Figure 3 of Stenchikov et al (JGR, 1991). It make no sense to average a large and a small forcing: that simply washes out the signal.*

P6, L18: Driscoll et all. (2012) adopted this methodology from Stenchikov et al. (2006).

*Thank you for pointing this out. We have now added the Stenchikov et al (2006) paper.*

P7, L8–9: The shortwave (SW) radiative forcing in three chosen models differs by 50%. There is much more differences in SW and Longwave (LW) aerosol absorption.

*Intermodel differences are not uncommon, and we have noted then.*

P7, L13–15: The models three times overestimate the equatorial lower stratospheric heating caused by volcanic aerosols. This is the main forcing of the stratosphere-troposphere dynamic interaction. There is something wrong here.

*Yes, we are agreed. This is a well know bias in many of the current-generation models. However, the reviewer will agree that the models we have analyzed are no more biased than the ones in Stenchikov et al (2006), Driscoll et al (2012), Robock & Zambri (2016) and many other studies.*

*More importantly: this model biases makes our argument stronger! Even with an over-estimated equatorial lower stratospheric heating caused by volcanic aerosols, our models (and those of the other recent studies) show no statistically significant surface warming. Had the models not overestimated the stratospheric aerosol heating the surface signal would be even smaller. We have noted this in the revised version of the paper (on page 7, lines 17–19).*

P7, L20: "With this IN mind"

*Thank you. We have corrected this.*

P7, L32–35: I think the correct question to ask is whether models are able to correctly reproduce the probability distributions of the Arctic oscillation (AO) responses to volcanic forcing. But for this one has to extract multiple cases from observations and to construct the observed probability distribution. It is not doable with only one post-eruption season considered.

*We politely disagree. We believe that one can indeed use "only one post-eruption season" provided on has large esembles of runs available. This is what we did.*

P9, L10: Planetary wave reflection is the threshold process (Charney and Drazin, 1961) when small changes matter.

> *We do not question the Charney and Drazin (1961) result, which is based on small-amplitude linear theory in a highly idealized configuration. What we question is how relevant that particular threshold behavior is to the problem at hand. The claim that a mere 1-2 m/s acceleration of the polar vortex from volcanic aerosol heating has a major impact on planetary wave propagation is purely speculative and, to the best of our knowledge, has yet to be demonstrated. For instance, one would first need to show that the climatological conditions are found to be very near the wave propagation threshold, so that a tiny wind perturbation is able to make the system cross that threshold. Then one would have to show that the linear approximations are actually valid. And so on. We are not aware of studies which have carefully performed such work.*

P9, L14: The wind variability coming from SSW is not relevant to the process. As soon as polar vortex zonal wind weakens below the threshold, planetary waves can propagate upward nonlinearly weakening the polar vortex. So, the amplitude of wind changes below the threshold, no matter how large it is, does not count. Sampling has to focus on the strong vortex cases for this purpose.

> *See the answer to our previous point, and also the answer to the other referee.*

P10, L17–18: Exactly, the winter of 1991/92 is not suitable to study the forced stratosphere-troposphere dynamic interaction, as the positive phase of AO in the troposphere was caused by a different mechanism.

> *This point has been addressed above, in the "General Comments" section.*

P11, L2: You mean surface cooling/warming, not in the lower stratosphere. Please clarify.

> *Yes, at the surface. Thanks for pointing out this ambiguity. We have correct the text.*

P11, L5-8: You have to explain why do we see a positive AO anomaly climatologically after multiple volcanic eruptions. If this would be extremely rear events as in the models, then a positive AO anomaly would not be seen in observations.

> *First: we do not think we need to explain "why we see a positive AO anomaly climatologically after multiple volcanic eruptions". Our paper is about Mt Pinatubo and that anomaly was absent in the winter following the 1991 eruption, demonstrating ipso facto that it could not possibly have been responsible for the observed NH surface warming.*

> *Second: the evidence for "a positive AO anomaly climatologically after multiple volcanic eruptions" is not terribly robust (see, e.g. Wunderlich & Mitchell, ACP 2017).*

P11, L15–23: The strengthening of the polar vortex caused by the volcanic aerosols heating in the lower equatorial stratosphere is robust. This is the threshold process, so a weak strengthening matters. And it is unfair to apply wind variability in SSW to scale the increase in maximum wind.

*We have addressed this comment in several locations in the discussion above.*

P11, L25–35: The downward propagation mechanism was proved using climatological analysis (Baldwin and Dunkerton, 1999) and has to be challenged on this basis.

*We are not sure what the referee means. We are not challenging the fact that SSWs can affect the annular modes and produce surface anomalies. That result is robust. We are questioning the claims of the early papers on the NH warming following volcanic eruptions. Those papers reported significant surface responses because the models used were flawed (they lacked vertical resolution and stratospheric variability). The fact that significant surface responses are not seen in the more recent studies (which are based on much better models) supports our interpretation.*

P12, L25–30: ENSO definitely could affect surface temperatures, although Volcano-ENSO interaction is highly nonlinear. At least contribution of ENSO variability in the volcanic signal has to be removed properly, which was never done in this analysis. Another important mode of variability is QBO that was not considered and reported in this study. QBO plays an important role in stratospheric wave propagation and could directly affect polar vortex and shape the stratosphere-troposphere dynamic interaction.

*We agree with the referee: ENSO and the QBO do affect stratospheric wave propagation. We also hope the referee agrees with us: that ENSO and the QBO are part of the internal variability of the climate system. Therefore, if their influence needs to be removed in order to detect any putative surface response to the volcanic aerosols (should it be detectable at all) it means that the volcanic surface influence in the NH is clearly masked by natural variability. This is the key point of our paper.*

---

## Referee Report (RR1)

Georgiy Stenchikov
Second Review
of the manuscript "Northern Hemisphere continental Winter-warming following the 1991 Mt. Pinatubo eruption: Reconciling models and observations" by L. Polvani et al.

I believe the paper raises a legitimate question about the nature and reality of a Winter warming response to volcanic forcing, but, with all respect to the authors, I do not believe they present convincing arguments to support their results, at least in the way they formulated them.

General Comments

The evidence of the development of a positive NAO/AO (further referred as AO) anomaly or Winter warming, in response to explosive equatorial volcanic eruptions, was first reported in the 1990s, and is based on compositing multiple observed volcanic events (e.g., Robock and Mao, 1992; Fisher et al., 2007). However, the AR4 and AR5 models tend to produce a weaker ensemble mean Winter warming than in observation composites (Stenchikov et al., 2006; Driscoll et al., 2012). Therefore, the dilemma is whether models are deficient, or Winter warming is spurious. The authors claim they solve this puzzle based solely on model output and observations for only one Winter, following the 1991 Pinatubo eruption.

We know that up-to-date models generate large uncertainties in reproducing circulation changes (e.g., Deser et al., 2012; Shepherd, 2014). The AO response to volcanic forcing, real or not, is an interesting example of dynamic perturbation caused by imposed radiative forcing. According to (Deser et al., 2012; Shepherd, 2014) it is not surprising that the models cannot capture it well.

A positive AO anomaly, after a volcanic eruption, can be generated by a number of stratospheric and tropospheric mechanisms (Stenchikov et al., 2002; Stenchikov, 2016). The stratospheric mechanism involves the strengthening of a NH Polar Vortex, which is relatively well-reproduced by the models, in general, and in this study particularly, and the downward propagation of a signal (Baldwin and Dunkerton, 1999), which is not well-captured by the models (Stenchikov et al., 2006; Driscoll et al., 2012).

Due to high variability, an individual Winter warming event is difficult to identify empirically (e.g., stand-alone 1991/92 Winter warming is not statistically significant). The conventional approach to reconcile model results and observations is to match a simulated ensemble mean and a statistically significant composited observed anomaly. Robock and Mao (1992) and Fisher et al. (2007) have composited several post-eruption events to obtain a statistically significant AO response; Stenchikov et al. (2006) and Driscoll et al. (2012) have composited model outputs. They all have to composite eruptions of different magnitudes. I do not think it is an unforgivable sin, assuming that the authors in this study compare the responses in the models where the SW flux, reflected by volcanic aerosols, differs by 50%, which is probably more than the difference between the NH radiative forcing of El Chichon and the forcing of Pinatubo.

In the current study, the authors choose to compare the "climate-type" large model ensembles with only one observed event: the Winter-warming response in the first year after the 1991 Pinatubo eruption. The Winter warming in 1991/92 is not typical because it is not associated with the strong NH polar vortex, as in the most post-volcanic years in observation. The asymmetry between 1991/92 and 1992/93 Winters caused by different phases of QBO is discussed in details by Stenchikov et al. (2004), based on a "large" 24-member ensemble, and using 40-layer stratosphere-resolving model. In addition, a Central Pacific El Nino of 1991/92 contributed into peculiarity of the chosen case-study (Predybaylo et al., 2017; Dogar et al., 2017). In the current study, none of those factors (QBO, El Nino) are accounted for in the simulations or, alternatively, their effects were not removed from observations, which as discussed in (Kirchner et al., 1999; Santer et al., 2001; Lehner et al., 2016) is nevertheless important.

Obviously, a simulated ensemble average cannot match the one natural realization, which is not statistically significant, as it comprises both the forced response and the natural variability. The authors stated in their response to the reviewer:
"Our key finding, after analyzing three large ensembles, is that the observed warming falls well within the distribution of the model members. From this, we conclude that the models capture the observations."
Basically, the authors claim that, if the observed response for one season falls within the spread of model ensemble responses ( i.e., there are a few ensemble members that show Winter warming), this fact validates the model. This is an overstatement. If a model is valid, then the observed response has to fall within the spread of model responses, but the opposite is not necessarily correct.

In summary , the actual results of the study do not support the authors' ambitious claim, as stated in the paper. The authors show that the models perturbed by the Pinatubo-like radiative forcing, due to a strong variability in high NH latitudes of the troposphere and the stratosphere, did not produce a statistically significant positive AO anomaly. It remains unclear if this would be right for the real physical system, or if it is the result of the model or experimental setup deficiencies. A comparison with the observed anomaly, for only one Winter following the 1991 Pinatubo eruption, does not sound convincing to me.

The models themselves do not perfectly simulate the Pinatubo impact. They generate volcanic radiative forcing with at least 50% uncertainty and overheat the equatorial lower stratosphere almost twice the amount in comparison with observations. The effects of QBO and ENSO on the Winter warming of 1991/92 are supposed to diminish in the model ensemble mean, but are not removed from observations; the ensemble sizes, at least for WACCAM –(the only stratosphere-resolving model used for the analysis) are relatively small. Clearly,  the models presented in this study and the method itself have some significant drawbacks. The conclusions are overstated and are at odds with empirical reconstructions (Robock and Mao, 1992; Fisher et al., 2007; Wunderlich and Mitchell, 2017) that are simply verbally dismissed. The results of the study are incorrectly interpreted. The conclusions should be made consistent with the actual results of the study, before  submission for publication in ACP.

Specific Comments

P1, L5: This is an overstatement; the strengthening of the NH polar vortex is often reproduced in the model simulations.

P1, L8: Which climate model is highly accurate? What does this mean? How did you prove it?

P1, L20: I believe most of these effects were previously discussed.

P2, L24: This is an inaccurate statement. Winter warming is associated with a positive phase of AO, and could occur independently of a volcanic impact.

P5, L16: English et al. (2013) did not account for the aerosol radiative feedback.

P7, L18: with this **in** mind

P7, L25: Stenchikov et al., **2006**

P7, L25-26: The statement made by the authors is inaccurate. Stenchikov et al. (2006) and Driscoll et al. (2012) compared observed and simulated anomalies composited for a few eruptions since 1850. They found that the simulated composited Winter warming is weaker than in observations.

P7, L30-35: Probably most, if not all, AR4 and AR5 models have ensemble members showing 1991/92 Winter warming, and therefore satisfy this suggested weakened criterion.

P8, L16: I do not think the authors, in their experiments, can claim "that the models are perfectly capable of capturing the post-Pinatubo Winter anomalies in the NH", based on the fact that a few ensemble members do this. E.g., the ensemble members that demonstrate Winter warming might do it for wrong reasons, as the models do not account for some important factors such as the Easterly QBO phase, El Nino in the Winter of 1991/92; models overheat the lower stratosphere and have 50% uncertainty in radiative forcing.

P9, L6: It would be fair to mention that, for some ensemble members, the zonal wind anomaly exceeds 10 m/s.

P9, L10-18: One possible explanation would be that WACCM does not capture the propagation of AO from the stratosphere to the troposphere, as in observations (Baldwin and Durkenton, 1999). Was WACCM tested in this way?

The vertical propagation of planetary waves is a threshold process (Charney and Drazin, 1961), so even small zonal wind changes might matter. The exact value of a threshold velocity obtained

in (Charney and Drazin, 1961) might not be  perfectly right in the real world, as it was obtained for idealized conditions. But a fundamental conclusion that a planetary wave propagation process is threshold should hold.

P10, L15-19: It is a sampling problem with only one post-Pinatubo season chosen. Multiple cases have to be considered to judge which mechanism works more frequently.

P10, L22: addre**sss**

P11, L20-30: Not all previous studies were conducted using models with a poorly resolved stratosphere. Stenchikov et al. (2002, 2004) used the 40-level GFDL stratosphere resolving model.

**References**

Baldwin, M. and T. Dunkerton (1999), Propagation of the Arctic Oscillation from the stratosphere to the troposphere, J. Geopys. Res., V. 104, No. D24, pp 30,937-30,946.

Deser C., R. Knutti, S. Solomon, A. Phillips (2012), Communication of the role of natural variability in future North American climate, Nature Climate Change, 2, DOI: 10.1038/NCLIMATE1562.

Dogar, M., G. Stenchikov, S. Osipov, B Wyman, M. Zhao (2017), Sensitivity of the Regional Climate in the Middle East and North Africa to Volcanic Perturbations, J. Geophys. Res. Atmos., 122, doi:10.1002/2017JD026783.

Driscoll, S, A. Bozzo, L.J. Gray, A.Robock, G.Stenchikov (2012), CMIP5 Simulations of Climate Following Volcanic Eruptions, J. Geophys. Res, 117, D17105, 26 pp., doi:10.1029/2012JD017607.

Fischer, E. M., J. Luterbacher, E. Zorita, S. F. B. Tett, C. Casty, H. Wanner (2007), European climate response to tropical volcanic eruptions over the last half millennium, Geophys. Res. Lett., V. 34, L05707, doi:10.1029/2006GL027992.

Kirchner, I., G. Stenchikov, H. Graf, A. Robock, J. Antuña, (1999), Climate Model Simulation of Winter warming and Summer Cooling Following the 1991 Mount Pinatubo Volcanic Eruption, J. Geophys. Res., 104, 19,039-19,055

Lehner, F., A. P. Schurer, G. C. Hegerl, C. Deser, and T. L. Frölicher (2016), The importance of ENSO phase during volcanic eruptions for detection and attribution, Geophys. Res. Lett., 43, 2851–2858, doi:10.1002/2016GL067935

Predybaylo, E., G. Stenchikov, A. Wittenberg, F. Zeng (2017), Impact of a Pinatubo-Size Volcanic Eruption on ENSO, J. Geophys. Res. Atmos., 122, 925-947, doi:10.1002/2016JD025796.

Robock, A. and Mao, J. (1992), Winter warming from large volcanic eruptions, Geophysical Research Letters, 19, 2405–2408.

Santer, B. D., et al. (2001), Accounting for the effect of volcanoes and ENSO in comparisons of modeled and observed temperature trends, J. Geophys. Res., 106, 28,033–28,059, doi:10.1029/2000JD000189.

Shepherd, T. (2014), Atmospheric circulation as a source of uncertainty in climate change projections, Nature Geosciences, 7, 703-708, DOI: 10.1038/NGEO2253.

Stenchikov, G., A. Robock, V. Ramaswamy, M. D. Schwarzkopf, K. Hamilton, S. Ramachandran (2002), Arctic oscillation response to the 1991 Mount Pinatubo eruption: Effect of volcanic aerosols and ozone depletion, J. Geophys. Res., 107 (D24), 4803, doi:10.1029/2002JD002090.

Stenchikov, G., K. Hamilton, A. Robock, V. Ramaswamy, and M. D. Schwarzkopf (2004), Arctic Oscillation response to the 1991 Pinatubo Eruption in the SKYHI GCM with a realistic Quasi-Biennial Oscillation, J. Geophys. Res., 109, D03112, doi:10.1029/2003JD003699.

Stenchikov, G., K. Hamilton, R. J. Stouffer, A. Robock, V. Ramaswamy, B. Santer, and H.-F. Graf (2006), Arctic Oscillation response to Volcanic Eruptions in the IPCC AR4 Climate Models, J. Geophys. Res., 111, D07107, doi:10.1029/2005JD006286.

Stenchikov, G. (2016), The Role of Volcanic Activity in Climate and Global Change, in: T.M. Letcher (Ed.), Climate Change: Observed Impacts on Planet Earth, Second Edition, Elsevier, 2016, pp. 419–447, ISBN: 978-0-444-63524-2.

Wunderlich, F. and D. Mitchell (2017), Revisiting the observed surface climate response to large volcanic eruptions, Atmos. Chem. Phys., 17, 485-499, https://doi.org/10.5194/acp-17-485-2017, doi:10.5194/acp-17-485-2017.

---

## Referee Report (RR2)

The authors' careful consideration of the two first-round reviews is appreciated, and has resulted in a number of minor edits that improve the manuscript. This manuscript as it stands is suitable for publication. Should additional revisions be made, a few comments are given below.

**Remaining comments**

P9 L15–17: My disagreement remains regarding the SSW–Pinatubo comparison. First, SSW events are fundamentally different than a (hypothetical) vortex acceleration from Mt. Pinatubo—with regard to mechanism, sign of the wind anomaly, magnitude, and timescale—so information about SSWs cannot be extrapolated to eruptions. Second, even if such extrapolation were appropriate, a single SSW may not reach the surface (as the authors note), but on average they do; and the strength of that effect is what is in question regarding Mt. Pinatubo, or eruptions in general.

That said, the sentence is not a big deal either way; I just think the SSW–Pinatubo comparison slightly weakens the overall argument of the manuscript.

F5: The additional analysis for Figure 5 is much appreciated, and adds support to the manuscript's narrative. Inclusion of the low-top results in Figure S6 is appropriate given that they are supporting, if weaker, evidence for the conclusions of Figure 6.

P9 L24–35: I previously commented that "examining two individual ensemble members does not offer any insight into the mechanism." The paragraph is certainly supported by Figure 6, but it is not surprising (at least to this reviewer) given the variety of processes contributing to interannual variability. Although my preference would be to keep the discussion focused on the effect/mechanism in the ensemble average, readers familiar with the authors' approach (as in the papers they cited) may wish to see the information presented in this form.

---

## Referee Report (RR3)

The authors of this manuscript argue that the large tropical eruption of Mt. Pinatubo has had little impact on Northern Hemisphere stratospheric polar vortex strength and virtually no impact on European surface temperature. With the help of large ensembles, they show that internal variability is sufficient to explain the observed temperature response after Mt. Pinatubo and possibly as well as for other large tropical eruptions. The proposed stratospheric mechanism of how volcanic eruptions dynamically influence European winter temperatures is hence called in the question by the authors.

I think the manuscript is important and of great scientific interest as it will possibly intensify the discussion about the dynamic impact of large volcanic eruptions which has been taken for granted so far. The impact of internal variability has been too much neglected so far, in this sense the manuscript offers a new, quite drastic, perspective. After addressing the few points I have, the manuscript should be suitable for publication.

**1. General comments** (I refer to the revised version of the manuscript):

1. I feel that the authors are a little too overconfident with the conclusion that internal variability alone is sufficient to explain the surface temperature signal in the winter following tropical volcanic eruptions. There is a significant acceleration of the polar vortex of 3,5-5 m/s (see specific comment below) after Pinatubo in one particular climate model as well as in the CMIP5 ensemble (Bittner et al. 2016). I agree that even 5 m/s is small compared to SSW events (or a strong acceleration of the vortex) but even the mean acceleration might have an impact on surface climate (Kidston et al., 2015). Moreover, the change in the mean can very well represent a change of polar vortex variability, i.e. more/less SSW or more episodes of strong vortices, on smaller timescales which have been shown to have an impact on surface climate (Baldwin and Dunkerton, 1999, and many others). One would need to investigate on smaller (probably daily) timescales how the vortex changes after volcanic eruptions in a large ensemble. I am not aware such a study has been done yet and it is clearly not the scope of this manuscript, but I would ask the authors to be more careful in completely dismissing the possibility of a stratospheric influence.

2. That said, I very much agree with the authors that with the too few observations at hand one can and should be skeptical about the "stratospheric mechanism". It might well be that the comparatively small acceleration of the NH polar vortex after volcanic eruptions are completely dwarfed by the internal variability. However, quite some observational studies show an impact of volcanic eruptions on European climate. In addition to the already cited Fischer (2007) and Shindell (2004), Christiansen (J. Clim., 2007) reports a significantly positive NAO and AO signal in the first winter after major eruptions since Krakatau (1883). Graf et al. (Clim. Dyn., 2014) show that the surface temperature signal under strong polar vortices are very different after volcanic eruptions in contrast to volcanically undisturbed winters. They note, however, the strong influence of internal variability (ENSO and QBO) and the limitation of the small sample size which prevent conclusive statements about mechanisms. Even if accounting for all the limitations of observations, especially if one goes back in time, I feel the authors are still too quick to dismiss the observational evidence. Even if Fischer (2007) reports a stronger surface influence of volcanic eruption in the second post-year eruption, it is very well possible that volcanic eruptions are partly responsible. Yes, I agree that averaging different eruptions strength can be problematic (as indicated in the manuscripts' discussion).

However, I'd rather argue that even if one has to average many eruptions (we will never get completely comparable Pinatubo eruptions in nature) and they seem to agree on some form of continental winter warming, there is likely to be a causal, physical connection. Of course, the volcanic influence is at least strongly modified by internal variability (as mentioned in the manuscript P2, LL21-25 as "perplexing fact"), but it is possible that a still unknown process is at work. Even if the stratospheric mechanism might not be as important as always assumed (or not important at all), there might be a tropospheric mechanism, involving maybe the ocean with a much longer memory, which influence European climate. With so many observational evidences I think it is rather unlikely that "everything is internal variability", hence I would ask the authors to acknowledge this conflict (observational studies vs. "everything is internal variability") and at least discuss the possibility of a volcanic influence on European winter temperatures which climate models might not capture correctly.

**2. Specific comments:**

P2, LL21-25: As mentioned in my general comment, I do not find it "perplexing" at all that smaller eruptions as El Chichon show a larger surface response compared to Krakatau or Tambora. As the authors stress, internal variability plays a crucial role, hence the possibly volcanic forced signal might be strongly modified/superimposed by internal variability.

P3, LL13-16: I frankly do not understand where these numbers come from. Bittner et al. (2016) show in Figure 2a a polar vortex acceleration of close to 4 m/s (ensemble average) not 2 m/s to a Pinatubo forcing. And these 4 m/s is statistically different from the null hypothesis at 15(25) ensemble members at the 95%(99%) confidence level. So, 100 model runs are more than sufficient to establish that fact.

P9, LL12-13: Same issue here. Which makes the agreement to WACCM4 even "more excellent", but the number of ensemble members are more like 15-25.

P10. L15: here again.

P11., L18: and here.

Christiansen, B. (2008). Volcanic eruptions, large-scale modes in the Northern Hemisphere, and the El Niño–Southern Oscillation. *Journal of Climate*, *21*(5), 910-922.

Graf, H. F., Zanchettin, D., Timmreck, C., & Bittner, M. (2014). Observational constraints on the tropospheric and near-surface winter signature of the Northern Hemisphere stratospheric polar vortex. *Climate dynamics*, *43*(12), 3245-3266.

Kidston, J., Scaife, A. A., Hardiman, S. C., Mitchell, D. M., Butchart, N., Baldwin, M. P., & Gray, L. J. (2015). Stratospheric influence on tropospheric jet streams, storm tracks and surface weather. *Nature Geoscience*, *8*(6), 433.

---

## Author Response (AR2)

**REPLY TO REFEREE REPORT #1    (Dr. Georgiy Stenchikov)**

I believe the paper raises a legitimate question about the nature and reality of a Winter warming response to volcanic forcing, but, with all respect to the authors, I do not believe they present convincing arguments to support their results, at least in the way they formulated them.

> *We are sorry to have been unable to convince Dr. Stenchikov, but our results are consistent with nearly all previous studies which have analyzed recent state-of-the-art models. Such models show* **no surface winter high-latitude warming in the ensemble mean** *following volcanic eruptions, just as our three large ensembles. What is different here is the interpretation: earlier studies concluded that the models are flawed, while we conclude that the models are fine and that the volcanic response is swamped by natural variability.*

**General comments**

The evidence of the development of a positive NAO/AO (further referred as AO) anomaly or Winter warming, in response to explosive equatorial volcanic eruptions, was first reported in the 1990s, and is based on compositing multiple observed volcanic events (e.g., Robock and Mao, 1992; Fisher et al., 2007). However, the AR4 and AR5 models tend to produce a weaker ensemble mean Winter warming than in observation composites (Stenchikov et al., 2006; Driscoll et al., 2012). Therefore, the dilemma is whether models are deficient, or Winter warming is spurious. The authors claim they solve this puzzle based solely on model output and observations for only one Winter, following the 1991 Pinatubo eruption.

> *We do not claim that the post-Pinatubo winter warming was "spurious": surface warming was observed and it was real. Our claim is that it was not caused by the eruption.*

We know that up-to-date models generate large uncertainties in reproducing circulation changes (e.g., Deser et al., 2012; Shepherd, 2014). The AO response to volcanic forcing, real or not, is an interesting example of dynamic perturbation caused by imposed radiative forcing. According to (Deser et al., 2012; Shepherd, 2014) it is not surprising that the models cannot capture it well.

> *Deser et al. (2012) and Shepherd (2014) do not state that the models "cannot capture the AO response well". Their papers emphasize that any forced dynamical response will be difficult to establish due to the large variability. Our study is an exellent example of this.*

A positive AO anomaly, after a volcanic eruption, can be generated by a number of stratospheric and tropospheric mechanisms (Stenchikov et al., 2002; Stenchikov, 2016). The stratospheric mechanism involves the strengthening of a NH Polar Vortex, which is relatively well-reproduced by the models, in general, and in this study particularly, and the downward propagation of a signal (Baldwin and Dunkerton, 1999), which is not well-captured by the models (Stenchikov et al., 2006; Driscoll et al., 2012).

*We politely disagree. The downward propagation of signals in most models is well captured. The models we have analyzed (and most other recent state-of-the-art models) can well reproduce the surface impact of SSWs and of stratospheric ozone depletion, both of which occur by a similar mechanism to the (alleged) volcanic signal. That mechanism entails (1) a temperature gradient in the stratosphere which (2) causes an anomaly in the tropospheric annular modes resulting in (3) a surface temperature anomaly over Eurasia. The reason the ensemble-mean shows no warming is that the signal-to-noise ratio is tiny, as we have shown. The same conclusion was reached by Bittner et al. (2016). A polar vortex acceleration of few meters/second is too small to significantly affect the annular modes, and this is why the models show no statistically significant forced temperature response at the surface.*

Due to high variability, an individual Winter warming event is difficult to identify empirically (e.g., stand-alone 1991/92 Winter warming is not statistically significant). The conventional approach to reconcile model results and observations is to match a simulated ensemble mean and a statistically significant composited observed anomaly. Robock and Mao (1992) and Fisher et al. (2007) have composited several post-eruption events to obtain a statistically significant AO response; Stenchikov et al. (2006) and Driscoll et al. (2012) have composited model outputs. They all have to composite eruptions of different magnitudes. I do not think it is an unforgivable sin, assuming that the authors in this study compare the responses in the models where the SW flux, reflected by volcanic aerosols, differs by 50%, which is probably more than the difference between the NH radiative forcing of El Chichon and the forcing of Pinatubo.

*The "conventional approach" was, in our opion, erroneous. Compositing only a few eruptions (typically a dozen) of different magnitudes, and mixing high- and low-latitude eruptions and first and second winters – as done by Robock and Mao (1992) and Shindell et al. (2004) and a few others – yields mostly a noisy signal. A more careful analysis, the one of Fischer et al. (2007), chose only large low-latitude eruptions, and composited first and second winters separately: from this they found that the surface signal is stronger in second winter, a fact that is hard to reconcile with a stratospheric pathway. We have discussed the observational evidence in detail on page 13 of the revised manuscript.*

In the current study, the authors choose to compare the climate-type large model ensembles with only one observed event: the Winter-warming response in the first year after the 1991 Pinatubo eruption. The Winter warming in 1991/92 is not typical because it is not associated with the strong NH polar vortex, as in the most post-volcanic years in observation. The asymmetry between 1991/92 and 1992/93 Winters caused by different phases of QBO is discussed in details by Stenchikov et al. (2004), based on a large 24-member ensemble, and using 40-layer stratosphere-resolving model. In addition, a Central Pacific El Nino of 1991/92 contributed into peculiarity of the chosen case-study (Predybaylo et al., 2017; Dogar et al., 2017). In the current study, none of those factors (QBO, El Nino) are accounted for in the simulations or, alternatively, their effects were not removed from observations, which as discussed in (Kirchner et al., 1999; Santer et al., 2001; Lehner et al., 2016) is nevertheless important.

*First: whether typical or not, the eruption of Mt. Pinatubo is the poster-child for the (alleged) surface winter warming "caused" by low-latitude volcanic eruptions, and was*

*used by Alan Robock in his highly-cited 2002 paper in Science. It is also the best observed large low-latitude eruption, and hence it is very much worth our understanding.*

*Second: the confounding effect of the QBO or ENSO support our interpretation. The very fact that those modes of natural variability were able to swamp the forced response, even for large eruption, such one of Mt. Pinatubo in 1992, shows that the volcanic response at the surface is small compared to the interval variability. In addition, the fact that the polar vortex was actually weaker (not stronger) in the winter after that eruption should suffice to convince anyone that the stratospheric pathway was surely not operative that winter, as it requires a vortex acceleration to cause a surface warming. The observed surface warming could not possibly have been caused by a decelerated polar vortex.*

Obviously, a simulated ensemble average cannot match the one natural realization, which is not statistically significant, as it comprises both the forced response and the natural variability. The authors stated in their response to the reviewer: "Our key finding, after analyzing three large ensembles, is that the observed warming falls well within the distribution of the model members. From this, we conclude that the models capture the observations." Basically, the authors claim that, if the observed response for one season falls within the spread of model ensemble responses ( i.e., there are a few ensemble members that show Winter warming), this fact validates the model. This is an overstatement. If a model is valid, then the observed response has to fall within the spread of model responses, but the opposite is not necessarily correct.

*If the observations fall within the range of an ensemble of model runs, one can conclude that the model captures the observations. This is not an overstatement: it is a reasonable conclusion. For all three models analyzed here, the observations happened to fall well within the model ensembles. Hence the title of our paper: it conveys the message that there is no conflict between models and observations for the most recent Pinatubo eruption.*

In summary , the actual results of the study do not support the authors' ambitious claim, as stated in the paper. The authors show that the models perturbed by the Pinatubo-like radiative forcing, due to a strong variability in high NH latitudes of the troposphere and the stratosphere, did not produce a statistically significant positive AO anomaly. It remains unclear if this would be right for the real physical system, or if it is the result of the model or experimental setup deficiencies. A comparison with the observed anomaly, for only one Winter following the 1991 Pinatubo eruption, does not sound convincing to me.

The models themselves do not perfectly simulate the Pinatubo impact. They generate volcanic radiative forcing with at least 50% uncertainty and overheat the equatorial lower stratosphere almost twice the amount in comparison with observations. The effects of QBO and ENSO on the Winter warming of 1991/92 are supposed to diminish in the model ensemble mean, but are not removed from observations; the ensemble sizes, at least for WACCAM (the only stratosphere-resolving model used for the analysis) are relatively small. Clearly, the models presented in this study and the method itself have some significant drawbacks. The conclusions are overstated and are at odds with empirical reconstructions (Robock and Mao, 1992; Fisher et al., 2007; Wunderlich and Mitchell, 2017) that are simply verbally dismissed. The results of the study are incorrectly interpreted. The conclusions should be made consistent with the actual results of the study, before submission for publication in ACP.

*The referee is raising a lot of different issues here. We separate them for clarity.*

*First, the models we have analyzed are of the same quality that those employed by most previous studies. They are state-of-the-art CMIP-class coupled climate models. We dispute the referee's claim that "the models presented in this study ... have some significant drawbacks." There are no other models available. If this type of models cannot be used to study volcanic impacts, all previous modeling papers would have to be discarded.*

*Second, our methodology is very simple. We just contrast the observations to an ensemble of model runs. Most previous papers contrasted the observations to the model mean: that was the mistake, as it ignored internal variability, and led to a perceived discrepancy between models and observations. Our paper shows that there is no discrepancy.*

*Third, the overheating in the equatorial lower stratosphere in our models is very similar to the one in the CMIP3 and CMIP5 models previously analyzed by Stenchikov et al. (2006) and Driscoll et al. (2012), and many other papers. That fact is not a reason to disqualify our paper. But, most importantly: that bias greatly strengthens our claim. Even with an unrealistically strong volcanic forcing, models show no statistical significant surface warming. Without bias, the odds of a significant surface warming would be even lower.*

*Fourth, we dispute the referee's claim that "the conclusions are overstated and are at odds with empirical reconstructions".*

- *As for overstatement: our revised manuscript now includes over three pages of discussion, where we carefully consider many aspects of the problem, and set our findings in the context of previous work and other eruptions.*

- *As for being at odds with reconstructions: we have already noted the issues with studies of Robock and Mao (1992) and Fischer et al. (2007). We believe these issue make that observational evidence questionable. Finally, we fear the referee might have misunderstood the findings of Wunderlich and Mitchell (2017): so we will spell them out. After examining Krakatau (August 1883), Santa Maria (October 1902), Agung (March 1963), El Chichon (April 1982) and Pinatubo (June 1991) with 9 reanalysis datasets (ERA-Interim, ERA-40, JRA-25, JRA-55, MERRA, NCEP-R1, NCEP-CFSR, NCEP-R2 and NOAA-20CR), the authors state (and we quote):*

  > Therefore we conclude that **we do not find a significant positive NAO response to volcanic eruptions** with taking the strongest five tropical eruption from the end of the 19th century until present (emphasis ours).

  *That study totally corroborates the key claim of our paper: it is not at odds with it.*

*Fifth and last: the conclusions of our paper are completely "consistent" with the results of our study with large ensembles. This is confirmed by the other two referees, who find our results novel and compeling, and support publication of our paper. Moreover, none of the issues raised above by referee argue that we are not being "consistent". So, we do not understand what "consistency" is being alluded in the last sentence.*

**Specific Comments**

P1, L5: This is an overstatement; the strengthening of the NH polar vortex is often reproduced in the model simulations.

*The strenghthening of the polar vortex was reported in the early studies which used models with very few vertical levels: it was a highly unrealistic feature, resulting from the lack of stratospheric variability in those early low-resolution model (as we discuss in the paper). Most recent models, e.g. the CMIP3 and CMIP5 models, show very little strengthening of the polar vortex. Bittner et al. (2016) and our own results here with WACCM, suggest that 15-20 members are needed to see a small strenghtening. We have, nonetheless, qualified this statement in the revised version.*

P1, L8: Which climate model is highly accurate? What does this mean? How did you prove it?

*We do not understand this comment; the referee perhaps misunderstood what we are saying. The word "accurate" was used as an hypothetical in that sentence.*

P1, L20: I believe most of these effects were previously discussed.

*We are not sure what "effects" the referee is alluding to. The word "effects" does not appear in that sentence, or that paragraph.*

P2, L24: This is an inaccurate statement. Winter warming is associated with a positive phase of AO, and could occur independently of a volcanic impact.

*We do not doubt that winter warming can be "associated with a positive phase of AO, and could occur independently of a volcanic impact." Our sentence is questioning the claim that volcanic eruptions would be an important driver of the warming. We make no mention of the AO in that sentence.*

P5, L16: English et al. (2013) did not account for the aerosol radiative feedback.

*We do not know what paper is being referred to: English et al. (20013) is not cited in our mascript, nor is it listed in the referee's comments.*

P7, L18: with this in mind

*We have corrected this typo.*

P7, L25: Stenchikov et al., 2006

*We are grateful for this correction, and now cite the 2006 paper.*

P7, L25-26: The statement made by the authors is inaccurate. Stenchikov et al. (2006) and Driscoll et al. (2012) compared observed and simulated anomalies composited for a few eruptions since 1850. They found that the simulated composited Winter warming is weaker than in observations.

> *We politely disagree with the referee. Driscoll et al. (2012) did not find the warming to be weaker: they found it to be* **statistically insignificant**. *Specifically, in the conclusion of that paper they state:*
>
> > *Disappointingly, we found that again, as with Stenchikov et al. (2006), despite relatively consistent post volcanic radiative changes,* **none of the models manage to simulate** *a sufficiently strong dynamical response.*
>
> *And again:*
>
> > *It is unclear why* **models fails to simulate the dynamics** *following volcanic eruptions.*

P7, L30-35: Probably most, if not all, AR4 and AR5 models have ensemble members showing 1991/92 Winter warming, and therefore satisfy this suggested weakened criterion.

> *We are agreed with the referee. We have no doubt that "most, if not all, AR4 and AR5 models have ensemble members showing 1991/92 Winter warming". We also don't doubt that as many ensemble members of those models show 1991/92 winter cooling. This is why the ensemble means shows nothing. We never claimed that the AR4 or AR5 models are flawed; they are similar to the models used in our study. Our claim is that the ensemble mean should not be compared to the observations. It is the entire distribution of the ensemble that needs to be compared to the observations.*

P8, L16: I do not think the authors, in their experiments, can claim that the models are perfectly capable of capturing the post-Pinatubo Winter anomalies in the NH, based on the fact that a few ensemble members do this. E.g., the ensemble members that demonstrate Winter warming might do it for wrong reasons, as the models do not account for some important factors such as the Easterly QBO phase, El Nino in the Winter of 1991/92; models overheat the lower stratosphere and have 50% uncertainty in radiative forcing.

> *Since many (not just a few) members show surface warming, we conclude that the models are capturing the observations. We see nothing wrong with our conclusion.*
>
> *We do not understand what the referee means with the expression "members that demonstrate winter warming might do it for wrong reason". What would be the "right reason" for the warming? Might the referee be implying the warming needs to caused by the volcanic eruption to be "correct"? If so, is the referee saying that warming due to internal variability is somehow "incorrect"? We are not sure what the referee means.*

*(We have already addressed the issue of model biases in lower stratospheric heating.)*

*As for the QBO and ENSO: they are indeed accounted for in our models, as they are part of the natural variability that is simulated by these models (although only WACCM has a QBO, as the other two models are low-top).*

P9, L6: It would be fair to mention that, for some ensemble members, the zonal wind anomaly exceeds 10 m/s.

*We see no need to mention this fact. We show the wind anomalies for all members of our ensembles, and for many members the zonal wind anomaly is actually* negative. *In any case, we do not understand why mentioning that fact would be "fair".*

P9, L10-18: One possible explanation would be that WACCM does not capture the propagation of AO from the stratosphere to the troposphere, as in observations (Baldwin and Durkenton, 1999). Was WACCM tested in this way?

*The WACCM model is stratosphere-resolving model that has been used for years to study stratosphere-troposphere dynamical coupling. It simulates a strong response of the tropospheric annular model to stratospheric ozone depletion, which causes a temperature gradient in the lower stratosphere not unlike the gradient caused by low-latitude volcanic eruptions (see, e.g. Neely et al, 2014). The WACCM model also has excellent stratospheric variability, e.g. the frequency of SSWs shown in Fig.3 of Marsh et al. (2013) compares very well with observations. It also simulated a highly relastic downward propagation of the annual mode signal, as shown in Fig. 4 of Marsh et al. (2013).*

The vertical propagation of planetary waves is a threshold process (Charney and Drazin, 1961), so even small zonal wind changes might matter. The exact value of a threshold velocity obtained in (Charney and Drazin, 1961) might not be perfectly right in the real world, as it was obtained for idealized conditions. But a fundamental conclusion that a planetary wave propagation process is threshold should hold.

*We have already addressed this issue in our previous reply. It is not relavant here.*

P10, L15-19: It is a sampling problem with only one post-Pinatubo season chosen. Multiple cases have to be considered to judge which mechanism works more frequently.

*We are not sure what the referee means by "multiple cases". We examine 13, 42, and 50 members across three ensemble. Driscoll et al. (2016) examined a dozen models with several members each. Bittner (2015) examined one model with 100 members.* **In all these instances the surface warming forced by the volcanic eruptions was not statistically significant***. We do not see why more cases are needed.*

P10, L22: addresss

*The typo has been corrected.*

P11, L20-30: Not all previous studies were conducted using models with a poorly resolved stratosphere. Stenchikov et al. (2002, 2004) used the 40-level GFDL stratosphere resolving model.

*We have qualified that sentence, which now reads "most of those early models simply lacked a good representation of the stratosphere..."*

**REPLY TO REFEREE REPORT #2**

The authors careful consideration of the two first-round reviews is appreciated, and has resulted in a number of minor edits that improve the manuscript. This manuscript as it stands is suitable for publication. Should additional revisions be made, a few comments are given below.

*We thank the referee for the careful reading and constructive comments.*

**Remaining comments**

P9 L1517: My disagreement remains regarding the SSWPinatubo comparison. First, SSW events are fundamentally different than a (hypothetical) vortex acceleration from Mt. Pinatubo – with regard to mechanism, sign of the wind anomaly, magnitude, and timescale – so information about SSWs cannot be extrapolated to eruptions. Second, even if such extrapolation were appropriate, a single SSW may not reach the surface (as the authors note), but on average they do; and the strength of that effect is what is in question regarding Mt. Pinatubo, or eruptions in general. That said, the sentence is not a big deal either way; I just think the SSW-Pinatubo comparison slightly weakens the overall argument of the manuscript.

*We politely disagree with the reviewer. While the time scales of SSWs may indeed be shorter than any possible vortex acceleration accompanying Mt. Pinatubo, stratospheric ozone depletion over the South Pole has similar time scale (of the order of several months) and it similarly affects the polar vortex and the annular modes. To the best of our knowledge, no one has claimed that the mechanism via which ozone depletion impacts the tropospheric annular modes is any different the mechanism associated with SSWs. Thus we see no reason why volcanic eruptions should be any different: they simply create a temperature gradient in the lower stratosphere, and the tropospheric annular modes respond accordingly. There is nothing special about eruptions. We think the analogy is entirely appropriate.*

F5: The additional analysis for Figure 5 is much appreciated, and adds support to the manuscripts narrative. Inclusion of the low-top results in Figure S6 is appropriate given that they are supporting, if weaker, evidence for the conclusions of Figure 6.

*We are agreed. And thank you for the suggestion: it improved our paper.*

P9 L2435: I previously commented that examining two individual ensemble members does not offer any insight into the mechanism. The paragraph is certainly supported by Figure 6, but it is not surprising (at least to this reviewer) given the variety of processes contributing to interannual variability. Although my preference would be to keep the discussion focused on the effect/mechanism in the ensemble average, readers familiar with the authors approach (as in the papers they cited) may wish to see the information presented in this form.

*Thank you. Showing that a large surface cooling can easily follow a strong eruption should be an eye-opener to many readers who have, until recently, only seen ensemble-means of model output (no paper to date has actually shown individual model runs).*

**REPLY TO REFEREE REPORT #3**

The authors of this manuscript argue that the large tropical eruption of Mt. Pinatubo has had little impact on Northern Hemisphere stratospheric polar vortex strength and virtually no impact on European surface temperature. With the help of large ensembles, they show that internal variability is sufficient to explain the observed temperature response after Mt. Pinatubo and possibly as well as for other large tropical eruptions. The proposed stratospheric mechanism of how volcanic eruptions dynamically influence European winter temperatures is hence called in the question by the authors.

I think the manuscript is important and of great scientific interest as it will possibly intensify the discussion about the dynamic impact of large volcanic eruptions which has been taken for granted so far. The impact of internal variability has been too much neglected so far, in this sense the manuscript offers a new, quite drastic, perspective. After addressing the few points I have, the manuscript should be suitable for publication.

> *We are grateful to the referee for taking the time to carefully read our manuscript, and for providing a thoughtful and helpful reply. We have addressed in detail the points he raises, and hope the manuscript will now be acceptable for publication.*

**1. General comments**

1. I feel that the authors are a little too overconfident with the conclusion that internal variability alone is sufficient to explain the surface temperature signal in the winter following tropical volcanic eruptions. There is a significant acceleration of the polar vortex of 3,5-5 m/s (see specific comment below) after Pinatubo in one particular climate model as well as in the CMIP5 ensemble (Bittner et al. 2016). I agree that even 5 m/s is small compared to SSW events (or a strong acceleration of the vortex) but even the mean acceleration might have an impact on surface climate (Kidston et al., 2015). Moreover, the change in the mean can very well represent a change of polar vortex variability, i.e. more/less SSW or more episodes of strong vortices, on smaller timescales which have been shown to have an impact on surface climate (Baldwin and Dunkerton, 1999, and many others). One would need to investigate on smaller (probably daily) timescales how the vortex changes after volcanic eruptions in a large ensemble. I am not aware such a study has been done yet and it is clearly not the scope of this manuscript, but I would ask the authors to be more careful in completely dismissing the possibility of a stratospheric influence.

> *That internal variability is sufficient to explain the surface temperature signal at the surface is a conclusion that is strongly suggested by our large ensemble results. This is exemplified in Figure 3, which shows how individual ensemble members can show equally large cooling as warming. Figures 3 and 4 show that the models' forced responses at the surface are zero, but that the observations are consistent with the models' simulation of internal variability. The same interpretation can be drawn from existing literature e.g. see the results from the CMIP5 ensemble in Figures 4 and 6 of Driscoll et al. [2012]. We are simply saying that if we believe that CMIP-class models are not fundamentally flawed in their representation of internal variability (and we are aware of no paper claiming and demonstrating that they are), there is no need to invoke a forced response (through the stratospheric pathway, or any other pathway, in fact) in order to explain the observed warming.*

*We question the stratospheric pathway by simply noting that the stratospheric polar vortex in the winter following the Pinatubo eruption was actually weaker, not stronger, then in the climatology (see our Figure 7d).*

*Finally, following the referee's suggestion, we have added a very detailed discussion (on page 13), an mention the possibility of other possible pathways for volcanic signals.*

2.  That said, I very much agree with the authors that with the too few observations at hand one can and should be skeptical about the stratospheric mechanism. It might well be that the comparatively small acceleration of the NH polar vortex after volcanic eruptions are completely dwarfed by the internal variability. However, quite some observational studies show an impact of volcanic eruptions on European climate. In addition to the already cited Fischer (2007) and Shindell (2004), Christiansen (J. Clim., 2007) reports a significantly positive NAO and AO signal in the first winter after major eruptions since Krakatau (1883). Graf et al. (Clim. Dyn., 2014) show that the surface temperature signal under strong polar vortices are very different after volcanic eruptions in contrast to volcanically undisturbed winters. They note, however, the strong influence of internal variability (ENSO and QBO) and the limitation of the small sample size which prevent conclusive statements about mechanisms. Even if accounting for all the limitations of observations, especially if one goes back in time, I feel the authors are still too quick to dismiss the observational evidence. Even if Fischer (2007) reports a stronger surface influence of volcanic eruption in the second post-year eruption, it is very well possible that volcanic eruptions are partly responsible. Yes, I agree that averaging different eruptions strength can be problematic (as indicated in the manuscripts' discussion).

However, I'd rather argue that even if one has to average many eruptions (we will never get completely comparable Pinatubo eruptions in nature) and they seem to agree on some form of continental winter warming, there is likely to be a causal, physical connection. Of course, the volcanic influence is at least strongly modified by internal variability (as mentioned in the manuscript P2, LL21-25 as perplexing fact), but it is possible that a still unknown process is at work. Even if the stratospheric mechanism might not be as important as always assumed (or not important at all), there might be a tropospheric mechanism, involving maybe the ocean with a much longer memory, which influence European climate. With so many observational evidences I think it is rather unlikely that everything is internal variability, hence I would ask the authors to acknowledge this conflict (observational studies vs. everything is internal variability) and at least discuss the possibility of a volcanic influence on European winter temperatures which climate models might not capture correctly.

*The reason for our skepticism of the observational evidence is that, leaving aside the issue of whether one ought to average eruptions of different magnitudes, the number of available eruptions is very small. Robock and Mao (1992) analyzed 12, but only 6 where in the tropics (equatorwards of 30°). Similarly Shindell et al (2004) analyzed 18 but, again, only 12 where at low latitudes. We wonder as to the consequences of mixing low and high latitude eruptions. Fisher et al (2007) averaged 15 low-latitude eruptions, but that number is still tiny in the context of the Bittner et al (2016) results: recall they they see no surface warming in winter even with 100 eruptions. In any case, we have now added (on page 13) a much more detailed discussion of the observational evidence, as suggested by the referee.*

**2. Specific comments**

P2, LL21-25: As mentioned in my general comment, I do not find it perplexing at all that smaller eruptions as El Chichon show a larger surface response compared to Krakatau or Tambora. As the authors stress, internal variability plays a crucial role, hence the possibly volcanic forced signal might be strongly modified/superimposed by internal variability.

> *We use the word 'perplexing' to alert the reader that a smaller warming following a larger eruption should immediately raise doubts as to whether the warming is caused by the eruption (which is what many papers have tacitly assumed, ignoring internal variability). We hope to have clarified this in the text.*

P3, LL13-16: I frankly do not understand where these numbers come from. Bittner et al. (2016) show in Figure 2a a polar vortex acceleration of close to 4 m/s (ensemble average) not 2 m/s to a Pinatubo forcing. And these 4 m/s is statistically different from the null hypothesis at 15(25) ensemble members at the 95%(99%) confidence level. So, 100 model runs are more than sufficient to establish that fact.

> *We are very grateful to the reviewer for flagging our incorrect reading of Bittner et al. (2016). We have corrected this here and in the two other instances. Indeed they state (1) that the polar vortex acceleration is 3-4 m/s following Pinatubo, and (2) that 15-20 model runs are needed to establish that fact at the 95% level.*

> *We also note, however, that while 100 members may be plenty to detect a statistically significant (albeit small) acceleration of the polar vortex, in his PhD thesis Bittner (2015, Fig 6.4) shows that even this large number of ensemble members is not sufficient to detect a statistically significant surface winter warming.*

P9, LL12-13: Same issue here. Which makes the agreement to WACCM4 even "more excellent", but the number of ensemble members are more like 15-25.

> *Thank you again. Fixed*

P10, L15: here again.

> *Fixed.*

P11, L18: and here.

> *Fixed.*

[revised manuscript text omitted]